# — The Flood Complex —
# Large-Scale Persistent Homology on Millions of Points

**Florian Graf**[1,†] ✉ **, Paolo Pellizzoni**[2,†] ✉ **, Martin Uray**[1,3]**, Stefan Huber**[3]**, Roland Kwitt**[1]

[1]University of Salzburg, Austria
[2]Max Planck Institute of Biochemistry, Germany
[3]Josef Ressel Centre for Intelligent and Secure Industrial Automation,
University of Applied Sciences, Salzburg, Austria

## Abstract

We consider the problem of computing Persistent Homology (PH) for large-scale Euclidean point cloud data, aimed at downstream machine learning tasks, where the exponential growth of the most widely-used Vietoris-Rips complex imposes serious computational limitations. Although more scalable alternatives such as the Alpha complex or sparse Rips approximations exist, they often still result in a prohibitively large number of simplices. This poses challenges in the complex construction and in the subsequent PH computation, prohibiting their use on large-scale point clouds. To mitigate these issues, we introduce the *Flood complex*, inspired by the advantages of the Alpha and Witness complex constructions. Informally, at a given filtration value $r \geq 0$, the Flood complex contains all simplices from a Delaunay triangulation of a small subset of a point cloud $X$ that are fully covered by the union of balls of radius $r$ emanating from $X$, a process we call *flooding*. Our construction allows for efficient PH computation, possesses several desirable theoretical properties, and is amenable to GPU parallelization. Scaling experiments on 3D point cloud data show that we can compute PH of up to dimension 2 on several millions of points. Importantly, when evaluating object classification performance on real-world and synthetic data, we provide evidence that this scaling capability is needed, especially if objects are geometrically or topologically complex, yielding performance superior to other PH-based methods and neural networks for point cloud data. Source code and datasets are available on ⃝: `https://github.com/plus-rkwitt/flooder`.

## 1   Introduction

Throughout the past years, topological data analysis (TDA) tools and, in particular, persistent homology (PH) [1, 21, 42, 54], have found widespread application in machine learning [39]. Applications range from graph classification [10, 28, 29], and time series forecasting [14, 51] to studying representations [2, 38, 48] and generalization of neural networks [7, 18], and developing novel regularizers or loss functions [13, 30]. The central pillar of most of these works is the power of PH to reveal and concisely summarize topological and geometrical information from a finite sample of points. In particular, within this pipeline, one first constructs a *simplicial complex* together with a *filtration* that encodes the underlying topological structure across scales. Once built, one then computes a stable summary called a *persistence barcode* or *persistence diagram*.

Importantly, however, the unfavorable computational complexity of the algorithmic pipeline for PH computation has, so far, largely limited the scope of topological approaches to studying connectivity

---

†equal contribution

39th Conference on Neural Information Processing Systems (NeurIPS 2025).

properties only. For this reason, efficient approaches to the computation of PH have been a goal for the field for some time, as emphasized in a recent position paper [39, Section 4].

When taking a closer look at how such simplicial complexes are typically built, we immediately identify potential scalability issues. In particular, when the input data is a point cloud, one often chooses the simplicial complex whose $k$-simplices are all subsets of cardinality $(k + 1)$. For example, the Vietoris-Rips and the Čech complex at filtration value $r$ consist of all subsets of the point cloud with a diameter less than $r$ and all subsets that can be enclosed in some ball of radius $r$, respectively. Hence, at high filtration values, their number of simplices becomes exponentially large, rendering memory consumption and persistent homology computation impractical for large point clouds. Simple mitigation strategies include computing the complex and homology only up to some degree, only up to some filtration value, or only on a *subsample* of the point cloud. More advanced methods, such as sparse approximations [44] and collapsing strategies [8], reduce the Vietoris-Rips complex to a smaller one with (approximately) equivalent topology. However, for (very) large point clouds, one may need to "sparsify" so aggressively that topological features and approximation guarantees no longer apply, while the resulting complex may *still* be too large for practical computation.

The natural way to avoid such scalability problems is to work with smaller complexes. Specifically, in low dimension (e.g., three as in our experiments), filtrations on the Delaunay triangulation of the point clouds are used, such as the Alpha complex [19], Delaunay-Čech complex [4] (both homotopy equivalent to Čech), or Delaunay-Rips [36] complex. However, in true large-scale settings, even the benign scaling of the Delaunay triangulation can be challenging, not in terms of memory consumption, but in terms of subsequent PH computation time of these filtered complexes, as the latter comes at worst-case cubic complexity (in the complex size) using the standard matrix reduction algorithm [21].

To further reduce the size of the complex, one can resort to *subsampling* strategies. In subsampling [9, 11], the key idea is to repeatedly draw small subsets of the full point cloud, execute the PH pipeline, and then aggregate the resulting persistence diagrams [9], or an appropriately vectorized representation of the latter [11]. Both approaches come with theoretical guarantees but, depending on the data, may require sufficiently large subsets, a large number of random draws, and a computationally expensive aggregation step (as in [9]). Crucially, once the subsamples from the point cloud are chosen, the complex construction is agnostic of the original data. Because of this, topological features that are smaller than the distance between samples are irreversibly lost.

The *Witness complex* construction [16], which is most similar to our approach, builds a simplicial complex in a data-driven manner on top of a reduced *landmark set*. In particular, simplices appear in the complex if they are witnessed, i.e., if there exists a single non-landmark point that satisfies a distance condition to the simplex. Similar to subsampling approaches, small topological features can be lost. Moreover, the construction of the Witness complex can be fragile, as it entails carefully controlling a distance cut-off to avoid truncating genuine topological features while taming the running time, which, in many cases, renders this type of construction impractical on large point clouds.

**Contribution(s).** We seek to mitigate the discussed scalability issues via the *Flood complex*, i.e., a new filtered simplicial complex for Euclidean point cloud data. Being built on top of a small subsample of the point cloud at hand, its size is orders of magnitude smaller not only than the Vietoris-Rips complex but also the Alpha complex, facilitating efficient PH computation. Moreover, as its simplices are endowed with filtration values that are tied to the *entire* point cloud, the topological approximation quality of the Flood complex is considerably better compared to subsampling strategies. The computation of the Flood complex is designed for execution on GPUs, enabling efficient exploitation of specialized hardware and software libraries frequently used in contemporary machine learning research. We provide (1) initial theoretical guarantees on the approximation quality of PH computed on the Flood complex, stability results, and guarantees for approximation steps. Further, we (2) demonstrate scalability to point cloud sizes that were previously impossible to process, and eventually (3) present strong empirical evidence that this scalability is needed and useful when seeking to classify geometrically complex 3D point cloud data.

## 2 Preliminaries

We study finite point sets $X$ in Euclidean space $\mathbb{R}^d$ in terms of the topology of the union of balls $X_r := \bigcup_{x \in X} B_r(x)$ at varying radii $r \geq 0$, with $B_r(x) := \{y \in \mathbb{R}^d : d(x, y) \leq r\}$ and metric $d(\cdot, \cdot)$. The topology of $X_r$ and, in particular, its homology, can be computed from a combinatorial object

called a simplicial complex. An (abstract) simplicial complex $\Sigma$ over a set $S$ is a collection of non-empty, finite subsets $\sigma \subset S$ such that for every non-empty subset $\tau \subset \sigma$ also $\tau \in \Sigma$. Moreover, $\sigma$ is called a $k$-simplex if it has cardinality $k + 1$, and a simplex $\tau \subset \sigma$ is called a *face* of $\sigma$.

As we study $X_r$ for different $r \geq 0$, we need a simplicial complex $\Sigma_r$ at each radius $r$. Analogously to $t \leq r$ implying $X_t \subset X_r$, we want $\Sigma_t \subset \Sigma_r$. A sequence of simplicial complexes $\{\Sigma_t : t \geq 0\}$ satisfying this property is called a *filtration* or *filtered simplicial complex*. In particular, a function $f : \Sigma_\infty \to \mathbb{R}$ that fulfills certain consistency properties induces a filtration via $\Sigma_t = f^{-1}((-\infty, t])$.

Informally, when increasing $r$, the shape of $X_r$ will gradually change from the points $X_0 = X$ themselves to $X_\infty = \mathbb{R}^d$. During this process, we can study $X_r$ by means of the homology groups $H_k$ of an associated simplicial complex $\Sigma_r$ and their evolution over $r$. Specifically, $H_0$ encodes connectivity information, $H_1$ information about loops, $H_2$ information about 3-dimensional voids (and subsequent homology groups encode higher dimensional topological information). As $r$ grows, these homological features appear and disappear, and we summarize this information in the form of a *persistence diagram* $\mathrm{dgm}_k(\Sigma)$, i.e., a multiset of tuples $(b, d) \in \mathbb{R}^2$, where $b$ denotes the minimal value $r$ such that a feature exists in $H_k(\Sigma_r)$ and $d$ denotes the maximal such value. Informally, the feature is born at time $b$ and dies at time $d$. Two persistence diagrams $D_1$ and $D_2$ can be compared via their *bottleneck distance* $d_B(D_1, D_2) := \inf_{\eta:D_1 \to D_2} \sup_{p \in D_1} \|p - \eta(p)\|_\infty$ where the infimum is taken over all bijections $\eta : D_1 \to D_2$, considering each diagonal point $(b, b)$ with infinite multiplicity. Importantly, if for two filtered complexes $\Sigma_r$ and $\Sigma'_r$, there exists $\epsilon > 0$ such that $\Sigma_{r-\epsilon} \subset \Sigma'_r \subset \Sigma_{r+\epsilon}$ for all $r \geq 0$, then $d_B(\mathrm{dgm}(\Sigma), \mathrm{dgm}(\Sigma')) \leq \epsilon$, see [15].

A particularly relevant family of simplicial complexes over point clouds $X$ are Alpha complexes [19] $\mathrm{Alpha}_r(X)$, which form a filtration of the Delaunay complex $\mathrm{Del}(X)$ on $X$. They are naturally embedded in $\mathbb{R}^d$ with each simplex $\sigma$ represented by its convex hull $\mathrm{conv}(\sigma) := \{\sum_{x \in \sigma} \lambda_x x : \sum_{x \in \sigma} \lambda_x = 1, \lambda_x \geq 0\}$. Denoting by $|\mathrm{Alpha}_r| := \bigcup_{\sigma \in \mathrm{Alpha}_r} \mathrm{conv}(\sigma) \subset \mathbb{R}^d$, we have that $|\mathrm{Alpha}_r| \subset X_r$ and that both are homotopy equivalent [19]. This relates back to the very first point, i.e., it guarantees that $X_r$ can be studied in terms of $\mathrm{Alpha}_r(X)$.

## 3 The Flood complex

As discussed, given a finite point cloud $X \subset \mathbb{R}^d$, ideally we would want to compute the PH of its (filtered) Alpha complex. The latter can be computed from a filtration of the Delaunay complex $\mathrm{Del}(X)$ which has, for $|X| = n$, at most $O(n^{\lfloor d/2 \rfloor})$ simplices (and thus much fewer than the $2^n - 1$ simplices of the Vietoris-Rips complex). However, the computation of PH will still be challenging if $n$ becomes very large. While, of course, one could compute the Alpha complex on only a subset $L \subset X$, one may lose valuable information when doing so. We therefore propose a filtration on $\mathrm{Del}(L)$ that is *aware* of the entire point cloud $X$.

Informally, at a given filtration value $r \geq 0$, our novel complex contains all simplices $\mathrm{Del}(L)$ that are fully covered by balls of radius $r$ emanating from $X$, a process we call *flooding*. We refer to the resulting complexes $\mathrm{Flood}_r(X, L) \subset \mathrm{Del}(L)$ as Flood complexes and, in alignment with the terminology of witness complexes, we will refer to $L$ as landmarks. We often select $L$ as a subset of $X$, but doing is not necessary, and the theoretical results in Section 3.2 do not require this assumption.

**Definition 1.** *For $r \geq 0$, the Flood complex $\mathrm{Flood}_r(X, L)$ at flood radius $r$ is the simplicial complex*

$$\mathrm{Flood}_r(X, L) = \left\{ \sigma \in \mathrm{Del}(L) : \mathrm{conv}(\sigma) \subset \bigcup_{x \in X} B_r(x) \right\} . \tag{1}$$

Whenever $0 \leq r \leq t$, we have $\mathrm{Flood}_r(X, L) \subset \mathrm{Flood}_t(X, L)$ from $X_r \subset X_t$, which yields a genuine filtration, i.e., a nested family $\{\mathrm{Flood}_r(X, L)\}_{r \in [0, \infty)}$ to which we refer to as the *filtered Flood complex*. By construction, if $L \subset X$, then all 0-simplices are already in the complex at time $r = 0$ and vice versa. For brevity, we will refer to the entire nested family of Flood complexes as $\mathrm{Flood}(X, L)$.

### 3.1 Intuition

Figure 1 illustrates a point cloud $X$, a subset $L \subset X$, and the construction of the filtered complexes $\mathrm{Alpha}(L)$ and $\mathrm{Flood}(X, L)$. We argue that $\mathrm{Alpha}_r(L)$, which is homotopy equivalent to the union of balls of radius $r$ centered on $L$, can be a poor representation of the topology of $X_r$, while the filtration function of the Flood complex makes it more aligned to the topology of the underlying data.

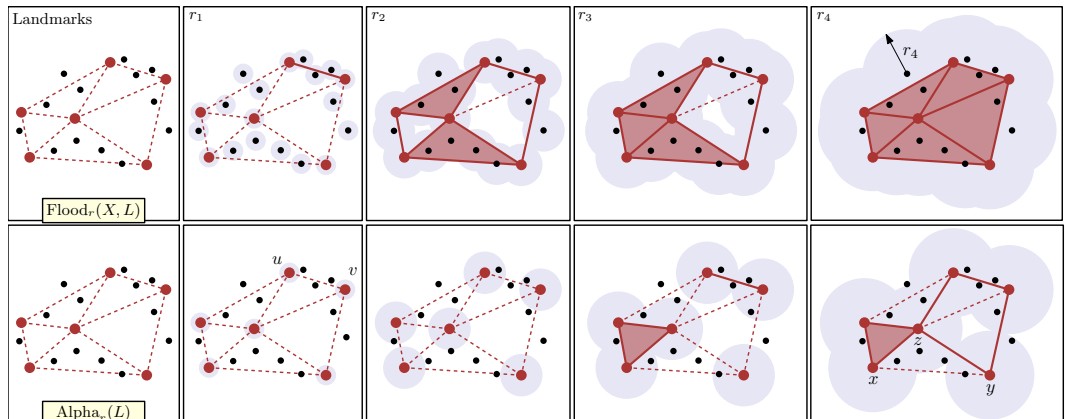

**Figure 1:** Schematic overview of the Flood complex $\text{Flood}_r(X, L)$ (**top**), the Alpha complex on a subsample $\text{Alpha}_r(L)$ (**bottom**), and their accordance with the union of balls $X_r$ at different radii $r$. The point cloud $X$ is marked by •, the landmarks $\subset X$ by •, and • identify the balls of radius $r$.

For example, the points $u, v \in X$ are in the same connected component of $X_{r_1}$, as illustrated in the second column of Figure 1. This is faithfully encoded in $\text{Flood}_{r_1}(X, L)$ which already contains the edge $\{u, v\}$, because $\text{conv}(\{u, v\}) \subset X_{r_1}$. However, in the Alpha complex, $u$ and $v$ will be in the same connected component much later, i.e., only after $r_3$. Similarly, at radius $r_2$, there are two cycles in $X_{r_2}$, which are recognized by $\text{Flood}_{r_2}(X, L)$ but not by $\text{Alpha}_{r_2}(L)$. Specifically, the rightmost cycle in $X_{r_2}$ is less persistent in the Alpha filtration, as it is born only much later in $\text{Alpha}(L)$ at filtration value $r_4$ and dies shortly afterwards. For the leftmost cycle this effect is even more pronounced. Similar considerations apply to higher order simplices, such as, for example, the triangle $\{x, y, z\}$, which is missing from $\text{Alpha}(L)$ even at filtration value $r_4$.

It is worth mentioning that since $\text{Flood}(X, L)$ is defined by a filtration on the Delaunay triangulation only on $L$ instead of $X$, there are natural limitations to its resolution. Most notably, every $k$-simplex in $\text{Del}(L)$ can detect at most one $k$-dimensional hole. Still, if the number of landmarks is sufficiently large and they are well-placed, then $\text{Flood}_r(X, L)$ will be able to capture the relevant homological information of $X_r$, as shown in the next subsection.

## 3.2 Theoretical results

The Flood complex satisfies certain beneficial properties (proofs can be found in Appendix C). First, it is stable with respect to the point cloud $X$. This means that if the point cloud $X$ is perturbed by a little, then the PH of $\text{Flood}(X, L)$ changes only a bit, quantified as follows.

**Theorem 2.** *The Flood complex is bottleneck stable with respect to its first argument, i.e., given $L, X, X' \subset \mathbb{R}^d$, it holds that $\forall i \in \mathbb{N}$*

$$d_B\left(\text{dgm}_i(\text{Flood}(X, L)), \text{dgm}_i(\text{Flood}(X', L))\right) \le d_H(X, X') \ . \tag{2}$$

A second desirable property satisfied by the Flood complex is that it recovers the topology of the union of balls $X_r$ when the landmarks $L$ are chosen as the whole point cloud $X$.

**Theorem 3.** *Let $X \subset \mathbb{R}^2$ be a finite subset of points in general position. Then, $|\text{Flood}_r(X, X)|$ is homotopy equivalent to $X_r$ for any $r \ge 0$.*

While, at the moment, our proof only covers the case $X \subset \mathbb{R}^2$, we conjecture that similar arguments work for higher dimensions, and we leave the proof for future work. Nevertheless, because homotopy equivalent spaces have the same homology, Theorem 3 directly implies that the persistent homology of $\text{Flood}(X, X)$ equals that of $\text{Alpha}(X)$. Moreover, in combination with Theorem 2 and the classic Hausdorff stability of Čech and Alpha complexes [15], we get the following approximation guarantee.

**Corollary 4.** *Let $L, X \subset \mathbb{R}^2$ be finite subsets of points in general position. Then, it holds that $\forall i \in \mathbb{N}$*

$$d_B\left(\text{dgm}_i(\text{Alpha}(X)), \text{dgm}_i(\text{Flood}(X, L))\right) \le 2d_H(X, L) \ . \tag{3}$$

Essentially, if we want $d_B(\text{dgm}_i(\text{Flood}(X, L)), \text{dgm}_i(\text{Alpha}(X))) \le 2\epsilon$, it suffices to select $L$ such that $d_H(X, L) \le \epsilon$. In particular, if $L \subset X$, finding the optimal landmark positions is just the metric

$k$-center problem, and greedy selection strategies, such as farthest point sampling (FPS), achieve a factor 2-approximation of the optimal covering radius [26]. Thus, $d_H(X, L) \leq \epsilon$ is achieved once $|L| \geq O(\epsilon^{-D})$ landmarks are selected, where $D$ is the intrinsic dimension of $X$. Similar guarantees hold with high probability if $X$ is a random sample and its first $|L|$ points are used as landmarks [34].

Moreover, Corollary 4 implies that the persistence diagrams of the Flood complex are stable with respect to the landmarks $L$, albeit with a multiplicative constant of 4. Still, in case $L$ is perturbed only slightly such that its Delaunay triangulation is preserved, then there is a direct proof that $d_B\left(\mathrm{dgm}_i(\mathrm{Flood}(X, L)), \mathrm{dgm}_i(\mathrm{Flood}(X, L'))\right) \leq d_H(L, L')$.

## 4 Computational aspects

We next discuss the computational aspects of constructing the Flood complex. Note that the set of simplices, i.e., a Delaunay triangulation of the landmark set $L$, can be computed in $O(|L|^{\lfloor d/2 \rfloor})$ [12], and efficient implementations can be found in libraries such as CGAL [45] or qhull [3].

### 4.1 Approximation

The definition of the Flood complex entails calculating the filtration value $f(\sigma)$ of each simplex $\sigma \in \mathrm{Del}(X)$. This filtration value $f(\sigma) = \max_{p \in \mathrm{conv}(\sigma)} \min_{x \in X} d(p, x)$ is the minimal radius $r$ such that $\mathrm{conv}(\sigma) \subset X_r$, or equivalently, the directed Hausdorff distance between $\mathrm{conv}(\sigma)$ and $X$. Since computing the directed Hausdorff distance is a nonconvex problem, finding $f(\sigma)$ would be inefficient. Hence, in our implementation, we replace each convex hull $\mathrm{conv}(\sigma)$ by a finite subset $P_\sigma \subset \mathrm{conv}(\sigma)$, resulting in the simplicial complex

$$\mathrm{Flood}_r(X, L, P) = \{\sigma \in \mathrm{Del}(L) \colon P_\tau \subset X_r \ \forall \tau \subset \sigma\} \ . \tag{4}$$

Specifically, $\sigma \in \mathrm{Flood}_r(X, L, P)$ iff $P_\sigma$ is flooded and all its faces $\tau \subset \sigma$ are in $\mathrm{Flood}_r(X, L, P)$. By construction, $\mathrm{Flood}_r(X, L) \subset \mathrm{Flood}_r(X, L, P)$, because $\mathrm{conv}(\sigma)$ being flooded implies that $P_\sigma$ is flooded. Moreover, if the (directed) Hausdorff distance between each pair $P_\sigma$ and $\mathrm{conv}(\sigma)$ satisfies $d_H(P, \mathrm{conv}(\sigma)) < \epsilon$, then $\mathrm{Flood}_r(X, L, P) \subset \mathrm{Flood}_{r+\epsilon}(X, L)$. We get the following guarantee.

**Theorem 5.** *Let $X, L \in \mathbb{R}^d$ and let $P = \{P_\sigma \colon \sigma \in \mathrm{Del}(L)\}$ with $P_\sigma \subset \mathrm{conv}(\sigma)$ for all $\sigma \in \mathrm{Del}(L)$. Then, it holds that $\forall i \in \mathbb{N}$*

$$d_B\left(\mathrm{dgm}_i(\mathrm{Flood}(X, L)), \mathrm{dgm}_i(\mathrm{Flood}(X, L, P))\right) \leq \max_{\sigma \in \mathrm{Del}(L)} d_H(P_\sigma, \mathrm{conv}(\sigma)) \ . \tag{5}$$

We can select the sets $P_\sigma$ in various ways, e.g., by explicitly enforcing a small Hausdorff distance to $\mathrm{conv}(\sigma)$, taking a random sample, or based on an evenly spaced grid of barycentric coordinates.

**Lemma 6.** *Given $m \in \mathbb{N}$ and a $k$-simplex $\sigma = \{v_0, \ldots, v_k\}$, let $P_\sigma = \{p = \sum_{i=0}^k \lambda_i v_i \colon \sum_{i=0}^k \lambda_i = 1, m\lambda_i \in \mathbb{N}\} \subset \mathrm{conv}(\sigma)$ be a grid of simplex points induced by evenly spaced barycentric coordinates. Then, it holds that*

$$d_H(P_\sigma, \mathrm{conv}(\sigma)) \leq \frac{1}{m} \sqrt{\sum_{i<j} \|v_i - v_j\|^2} \ .$$

Notably, the bound decays with $1/m$, where $m + 1$ is the number of grid points on each edge. The shape of the simplices enters in form of the square root term, which scales with $O(\mathrm{diam}(\sigma))$. A similar scaling behavior can be observed when $P_\sigma$ is selected randomly, see Lemma 9.

Thanks to the finite set $P_\sigma$, the optimization problem can be solved directly by iterating over the points in $P_\sigma$ and $X$. However, the number of points in $P_\sigma$ required for a good approximation is large, so this approach does not scale to large point clouds if implemented naively. As the problem entails finding the nearest neighbor in $X$ for each point in $P_\sigma$, it can be efficiently solved using a data structure such as a $k$-d tree [5] or a cover tree [6]. Building such a data structure can, however, incur significant computational overhead. To avoid this issue, we compute the directed Hausdorff distance using a custom GPU algorithm, paired with a principled *masking* procedure, which is presented below.

### 4.2 Masking and GPU acceleration for the flooding process

Since the filtration value $\max_{p \in P_\sigma} \min_{x \in X} d(p, x)$ of the simplex $\sigma$ depends only on a single pair of points $(p, x)$, we want to ignore points $x \in X$ that are far from $\sigma$. In fact, if $L \subset X$, then we can precompute for each simplex $\sigma$ a subset of $X$ that must be considered.

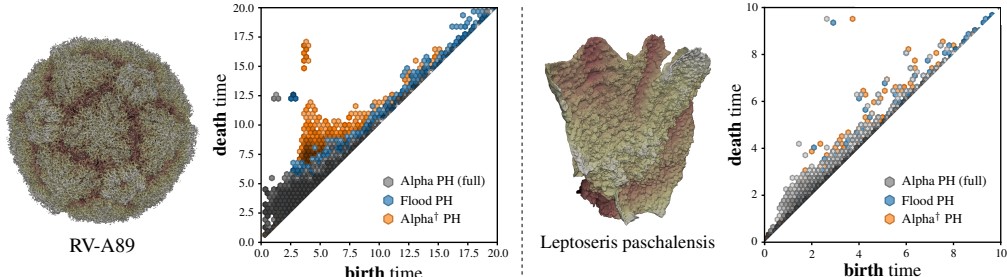

**Figure 2:** Exemplary hexbin plots of persistence diagrams of RV-A89 (**left**) and the Leptoseris paschalensis coral (**right**). Gray corresponds to Alpha PH of the full point cloud, blue to Flood PH with 10k landmarks and orange to Alpha† PH with 75k points. Point clouds are visualized via small spheres colored by distance to their bounding box center (for RV-A89) or by elevation (for the Leptoseris paschalensis coral). Best viewed in color.

**Lemma 7.** *Let $X \subset \mathbb{R}^d$ be a finite set in general position, and let $\sigma \subset X$ be a set of $k \leq d + 1$ points. If $c \in \mathbb{R}^d$ and $r > 0$ are such that $\sigma \subset B_r(c)$, i.e., such that $B_r(c)$ is an enclosing ball of $\sigma$, then*

$$f(\sigma) = \max_{p \in \text{conv}(\sigma)} \min\{d(p,x) : x \in X\} = \max_{p \in \text{conv}(\sigma)} \min\left\{d(p,x) : x \in X \cap B_{\sqrt{2}r}(c)\right\} . \quad (6)$$

In practice, we compute the enclosing balls $B_r(c)$ using a variation of Ritter's heuristic [41], i.e., we set $c$ as the center of the longest edge of $\sigma$ and use radius $\sqrt{2} \max_{i=1}^k d(c, v_i)$. This can be done in $O(d^2)$ time and is highly parallelizable on GPUs. In case $\sigma$ is itself an edge, it suffices to set the radius to half its diameter, i.e., without the factor $\sqrt{2}$.

Once centers and radii have been computed for all simplices $\sigma \in \text{Del}(L)$, checking whether the points in $X$ belong to $B_{\sqrt{2}r}(c)$ can be done with $|\text{Del}(L)| \cdot |X|$ distance evaluations, yielding a Boolean *masking* matrix of shape $|\text{Del}(L)| \times |X|$. Since the mask can be computed independently for each (simplex, point) pair, computation of the masking matrix can be efficiently parallelized. Eventually, the masking matrix together with the points in $X$ and the sets $P_\sigma$ are fed to a custom Triton [47] kernel, which computes the directed Hausdorff distance; for specifics of this kernel, see Appendix A.1.

## 5 Experiments

We validate the applicability and relevance of the Flood complex as follows: in Section 5.1, we use it to compute PH on point clouds for which existing approaches require an impractical amount of computational resources; in Section 5.2, we study the scalability of the Flood complex, and, in Section 5.3, we show that PH computed on the Flood complex (in short, Flood PH) improves predictions in downstream machine learning tasks compared to simpler approaches such as subsampling.

In our experiments, we often compare the Flood complex and its PH to the Alpha complex on the entire point cloud and to the Alpha complex on a subset of size selected so that runtime is similar to Flood PH at the given number of landmarks and discretization level of simplices. Specifically, we discretize to 30 points per edge in Sections 5.1 and 5.2 and 20 points per edge in Section 5.3. We denote the former by Alpha PH (full) and the latter by Alpha† PH. For PH computation and the construction of Alpha complexes, we use Gudhi [46].

### 5.1 Persistent homology on large-scale point clouds

First, we showcase the approximation capabilities of the Flood complex on two exemplary real-world point clouds from different scientific disciplines, one (12M points) based on a cryo-electron microscopy of a rhinovirus (RV-A89) from [49], the other (10M points) based on a 3D-scan of a Leptoseris paschalensis coral from the *Smithsonian 3D Digitization* initiative. Additional information about the point clouds can be found in Appendix A.4. We compute zero-, one-, and two-dimensional PH using the Alpha complex, which can be thought of as the *ground truth*. Notably, this requires more than 15 minutes of runtime per point cloud and more than 90 GB of RAM. We then compare its PH to that of the Flood complex and to that of the Alpha complex computed on a subsample.

A visual inspection of the persistence diagrams, see Figure 2, shows that Flood PH mitigates the shift of (birth, death)-tuples along the diagonal (that is characteristic for subsampling methods), resulting

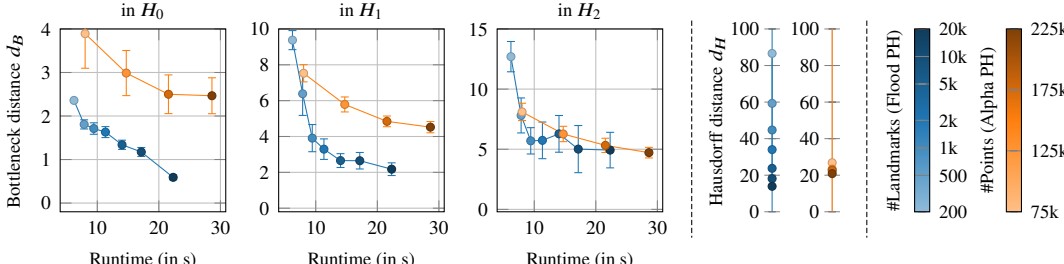

**Figure 3:** *Approximation quality* of Flood PH and Alpha PH on RV-A89. The **(left)** panel shows bottleneck distances to Alpha PH (full) in $H_0$, $H_1$ and $H_2$ when varying the number of landmarks for Flood PH and the subsample size for Alpha PH. The **(middle)** panel shows the Hausdorff distance between the full point cloud and the landmarks, resp., the points in the subsample. The **(right)** panel shows the color coding used in all the plots.

in a better approximation of birth and death times compared to Alpha PH. Moreover, the persistent $H_1$ features of the RV-A89 virus become scattered for Alpha† PH but remain close together for Flood PH.

For a quantitative comparison between Flood PH and Alpha PH on subsamples, we consider bottleneck distances to Alpha PH (full) on the RV-A89 point cloud. Results are presented in Figure 3; for results on the Leptoseris coral, we refer to Appendix B.2. As expected, the approximation quality of Flood PH improves with the number of landmarks. Notably, the randomness in the landmarks, caused by different starting points for FPS, mainly affects $H_2$. Beyond 500 landmarks, bottleneck distances are significantly smaller in dimensions 0 and 1, and similar in dimension 2, when comparing Flood PH to Alpha PH at the same runtime budget. A comparison between landmark selection using farthest point and uniform random sampling is reported in Appendix B.3.

## 5.2 Runtime and scalability

**Breakdown of runtimes**. In Table 1, we report the runtime share of different parts of Flood PH. Specifically, we consider one point cloud from the `swisscheese` dataset (1M points) and one from the RV-A89 (12M points), see Section 5.1. We compute filtration values for each simplex from a grid with 30 points per edge and use FPS for landmark selection (2k and 10k). Although the (relative) runtimes inevitably depend on the configuration of the Flood complex and the shape and size of the point clouds, several trends can be observed: the majority of time is spent computing filtration values, followed by

**Table 1:** Relative runtime breakdown (in %) of Flood PH on *one* point cloud from two datasets.

|  | `swisscheese` | RV-A89 | |
|---|---|---|---|
| #Points $\lvert X \rvert$ | 1M | 12M | 12M |
| #Landmarks $\lvert L \rvert$ | 2k | 2k | 10k |
| Landmark select. | 13.3 | 15.8 | 18.1 |
| Delaunay triang. | 6.0 | 0.8 | 2.9 |
| Masking | 3.7 | 3.2 | 7.7 |
| Filtration | 69.5 | 79.3 | 68.2 |
| PH computation | 3.0 | 0.2 | 1.0 |
| Other | 4.4 | 0.6 | 2.1 |

landmark selection and masking; the runtime of the remaining parts, i.e., triangulating the landmarks, PH computation and other overhead (e.g., computing grid points and enclosing balls of simplices) mainly depends on the number of landmarks. On RV-A89 and $\lvert L \rvert = 2k$, this overhead is negligible.

**Scalability**. We assess the scaling behavior of the Flood complex on point clouds constructed in the same manner as the `swisscheese` dataset. Specifically, we study its scaling wrt. (a) the number $\lvert X \rvert$ of points in the point cloud, (b) the number $\lvert L \rvert$ of landmarks, and (c) the ambient dimension. We always compare to Alpha PH computed from subsamples of the same size as $\lvert X \rvert$. As can be seen in Figure 4 (a), both methods exhibit an increase in runtime with $\lvert X \rvert$. However, while for $\lvert X \rvert < 10^4$, the Flood PH runtime is similar to Alpha PH (likely due to the overhead of landmark selection), for larger point clouds, Flood PH requires consistently one to two orders of magnitude less time than Alpha PH. Figure 4 (b) presents the runtime of Flood PH as a function of $\lvert L \rvert$ (with $\lvert X \rvert = 1M$ fixed). In general, we observe a linear scaling behavior that is aligned with the typically linear growth [22] in the number of simplices in the Delaunay triangulation of $L$. Finally, Figure 4 (c) shows the impact of the ambient dimensionality $d \in \{2, 3, 4, 5\}$ when keeping $\lvert X \rvert = 1M$ and $\lvert L \rvert = 1k$ fixed. Here, runtimes increase with $d$ for both methods, as the number of simplices in the Delaunay triangulations grows. Notably, Flood PH consistently remains at least one order of magnitude faster than Alpha PH.

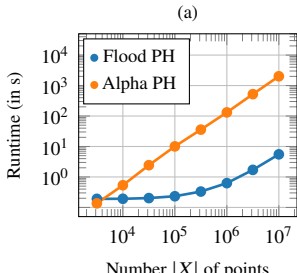
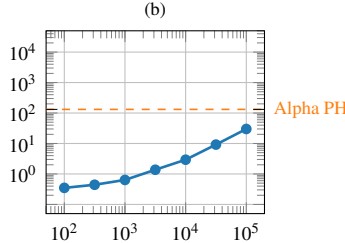
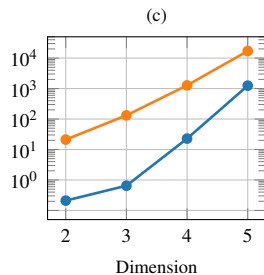

**Figure 4:** Runtime (in s) of Flood PH and Alpha PH for `swisscheese`-like point clouds: (a) in $\mathbb{R}^3$, varying the point cloud size $|X|$ with $|L| = 1k$ landmarks; (b) in $\mathbb{R}^3$, varying the number of landmarks $|L|$ with $|X| = 1M$ points; (c) varying the dimensionality with $|X| = 1M$ and $|L| = 1k$.

## 5.3 Object classification

We run classification experiments on five datasets, including three real-world datasets, i.e., `modelnet10`, `mcb-c` and the self-curated `corals`, which come in the form of surface meshes with a varying number of vertices and varying object "complexity". In terms of the latter, the number of vertices and triangles is a good proxy (especially for the CAD-based models in `mcb-c`), as, e.g., representing a helical geared motor requires a finer mesh than representing a taper pin. To create point clouds, we uniformly sample $|X|$ points from the mesh surfaces (as, e.g., done in [17]), which mitigates artifacts from irregularly spaced vertices. We also create two topologically challenging 3D synthetic datasets, i.e., `swisscheese` and `rocks`. Dataset details are provided in Appendix A.4. We use *ten* random 80/20% training/testing splits, with 10% of the training data reserved for validation.

**Parametrization of the Flood complex & classifier.** Unless otherwise stated, we select $|L| = 2k$ landmarks from $|X| = 1M$ points using FPS. To compute filtration values, we discretize each simplex based on an equally spaced grid of barycentric coordinates with 20 points per edge (resulting in 210 points per triangle and 1540 points per tetrahedron); cf. Section 4.1. We vectorize persistence diagrams ($H_0$, $H_1$ and $H_2$) using [27] with (exponential) structure elements, parametrized as follows: *locations* are set to 64 $k$-means++ centers, computed from the (birth, death) tuples of all diagrams in the training data, *scales* are chosen as in ATOL [43, Eq. (2)], and the vectorization's *stretch* parameter is set to either the one, five, or ten percent lifetime quantile (based on the validation data); this yields 64-dim. vectorizations per diagram which, upon concatenation, are fed as 192-dim. feature vectors to an LGBM [31] classifier (except for `corals`, where we use $\ell_1$-regularized logistic regression). Hyperparameters are tuned on the validation data using FLAML [50] and a time budget of 10 minutes.

**Baselines.** Our primary TDA baseline is Alpha[†] PH, i.e., persistent homology computed from a subsample of size chosen such that the runtime is similar to Flood PH, i.e. 1% of the `rocks` point clouds and otherwise 20k points . Persistence diagrams are processed in the same manner as described above. Importantly, we do *not* account for the vectorization time when choosing the subsample size for Alpha[†] PH, although persistence diagrams obtained from Alpha complexes can become very large and require substantially more time for vectorization than those obtained from Flood PH. We also compare to *averaged* persistence diagram vectorizations [11], denoted as Alpha PH (avg.), obtained from Alpha PH of *five* point cloud subsamples of accordingly reduced size. Unfortunately, methods such as Witness complexes [16], sparse Rips filtrations [44], or adaptive approximations [25], which might at first glance seem natural competitors to the Flood complex, proved to be infeasible on large-scale point clouds. For example, computing a lazy Witness complex with 100 landmarks and 1M witnesses required more than 20 minutes on our hardware (using the `Gudhi` [46] implementation).

Moreover, we compare to three neural network methods, designed for point cloud data: PointNet++ [40], PointMLP [35], and PVT [52] (with network width and depth as in the reference implementations). We train all models on point clouds subsampled to 2k points by minimizing cross-entropy (or MSE for the regression task on `rocks`) over 200 epochs using Adam [33] with a cosine annealing schedule and batch size 64. We select the initial learning rate, weight decay, and early stopping period based on the validation data. On the real-world datasets, we use random scaling and shifts as data augmentation.

**Evaluation metrics.** For classification tasks, we report the mean balanced accuracy over training/testing splits $\pm$ 1 standard deviation on all datasets (to account for class imbalances). For the regression task on `rocks`, we report the mean of mean squared errors (MSE) over splits $\pm$ 1 standard deviation.

**Table 2:** Classification results on **synthetic** data. *Runtime* (in s) is given per point cloud (averaged over all point clouds in a dataset); *Acc* denotes balanced accuracy (averaged over all ten splits). We do not list runtimes for neural networks (bottom part), as they are not directly comparable. Further, there is only one *Runtime* column for `rocks`, as the classification and regression task use the same persistence diagrams. ⏱ indicates that results cannot be computed within a 48-hour time budget (runtime for `rocks` is estimated).

| | swisscheese (2) | | rocks (2) | rocks (reg) | |
| | Acc ↑ | Runtime ↓ | Acc ↑ | MSE ↓ | Runtime ↓ |
|---|---|---|---|---|---|
| **Flood PH** | **0.98** ± 0.01 | 1.1 ± 0.1 | **0.88** ± 0.03 | ( **0.5** ± 0.2) × 1e-3 | 7.0 ± 2.9 |
| Alpha[†] PH | 0.85 ± 0.04 | 1.8 ± 0.0 | 0.78 ± 0.03 | ( 1.5 ± 0.3) × 1e-3 | 9.1 ± 4.2 |
| Alpha[†] PH (avg.) | 0.80 ± 0.02 | 0.9 ± 0.0 | 0.76 ± 0.04 | (20.9 ± 1.3) × 1e-3 | 6.9 ± 3.1 |
| Alpha PH (full) | 0.94 ± 0.02 | 134.3 ± 15.8 | ⏱ | ⏱ | 1490 ± 680 |
| PointNet++ [40] | 0.49 ± 0.03 | - | 0.54 ± 0.01 | (18.6 ± 2.8) × 1e-3 | - |
| PointMLP [35] | 0.51 ± 0.02 | - | 0.51 ± 0.03 | ( 8.8 ± 1.0) × 1e-3 | - |
| PVT [52] | 0.50 ± 0.02 | - | 0.51 ± 0.04 | (31.4 ± 3.4) × 1e-3 | - |

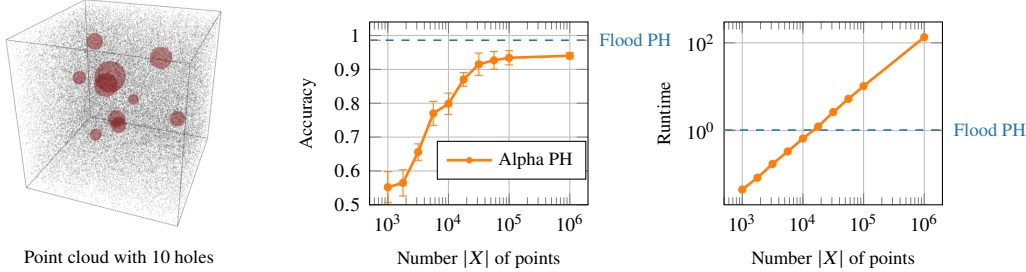

Point cloud with 10 holes

**Figure 5:** Comparison of classification of accuracy (on `swisscheese`) and runtime (in s) between Flood PH (2k landmarks) and Alpha PH when the latter has access to an increasing number of points in $X$. The leftmost panel shows an example of a `swisscheese` point cloud with 10 holes.

### 5.3.1 Synthetic data

**Swiss cheese**. We create a synthetic dataset of 3D point clouds by uniformly sampling 1M points in $[0, 5]^3$ and removing $k$ disjoint balls with random radii (in $[0.1, 0.5]$) and centers. We set $k \in \{10, 20\}$ and seek to distinguish point clouds by their number of voids (i.e., a *binary* problem). An example (with $k = 10$ voids highlighted in red) is shown in Figure 5 (left). The motivation for creating this dataset is to demonstrate that any approach based solely on subsampling the data for computational reasons (either in the context of computing PH, or for training neural network models) will perform poorly, as the class-specific topological characteristics (i.e., the voids in $H_2$) will be difficult to identify.

Table 2 confirms that all neural network baselines fail on this problem. Moreover, the classification performance of Alpha[†] PH is worse than that of Flood PH, most likely because Alpha[†] PH provides less information. To examine the latter, we compare Flood PH with Alpha PH on increasingly larger subsampled point clouds in Figure 5. As expected, the classification accuracy increases with the subsample size. However, surprisingly, it never matches the accuracy of Flood PH, but reaches its maximum of 94% at 1M points, i.e., at the entire point cloud. We hypothesize that the remaining gap to Flood PH can be attributed to the very large size of the resulting persistence diagrams, making it difficult to represent the relevant information in the vectorization. Furthermore, we observe an approximate linear increase in runtime for Alpha PH with increasing point cloud size, resulting in more than 24 hours runtime for computing PH on the entire dataset when all 1M points are used.

**Rocks**. Extending the ideas underlying the `swisscheese` data, we generate a more difficult dataset that mimics *porous materials*. Its point clouds contain up to 16M points and are extracted from Boolean voxel grids of size $256^3$, produced using two different generators (blobs and fractal noise) available as part of the `PoreSpy` library [23]. We evaluate classification accuracy wrt. the data generating method (i.e., a *binary* problem) and regression accuracy wrt. the surface area computed from the voxel grid. From Table 2, it is apparent that PH, and particularly Flood PH, excels on this dataset, achieving an average classification accuracy of 88%, whereas neural networks are only marginally better than random guessing. Similarly, on the regression task, Flood PH performs by far the best, with all neural networks yielding MSEs more than one magnitude larger than Flood PH.

**Table 3:** Classification results on **real world** data. *Runtime* (in s) is given per point cloud (averaged over all point clouds in a dataset); *Acc* denotes balanced accuracy (averaged over all ten splits). We do not list runtimes for neural networks (bottom part), as they are not directly comparable. ⊙ indicates that results cannot be computed within a 48-hour time budget.

| | corals (2) | | mcb-c (11) | | modelnet10 (10) | |
|---|---|---|---|---|---|---|
| | Acc ↑ | Runtime ↓ | Acc ↑ | Runtime ↓ | Acc ↑ | Runtime ↓ |
| **Flood PH** | **0.77** ± 0.09 | 1.8 ± 1.2 | **0.74** ± 0.02 | 1.5 ± 0.5 | 0.72 ± 0.02 | 0.8 ± 0.2 |
| Alpha$^\dagger$ PH | 0.58 ± 0.13 | 1.7 ± 0.3 | 0.66 ± 0.03 | 1.8 ± 0.2 | 0.64 ± 0.01 | 1.6 ± 0.1 |
| Alpha$^\dagger$ PH (avg.) | 0.52 ± 0.08 | 1.4 ± 0.1 | 0.69 ± 0.03 | 1.4 ± 0.1 | 0.66 ± 0.02 | 1.3 ± 0.1 |
| Alpha PH (full) | 0.44 ± 0.07 | 153.4 ± 16.8 | ⊙ | 119.1 ± 7.6 | ⊙ | 107.3 ± 5.9 |
| PointNet++ [40] | 0.53 ± 0.10 | - | **0.76** ± 0.02 | - | **0.93** ± 0.01 | - |
| PointMLP [35] | 0.59 ± 0.15 | - | **0.74** ± 0.03 | - | **0.93** ± 0.01 | - |
| PVT [52] | 0.56 ± 0.11 | - | **0.77** ± 0.03 | - | **0.94** ± 0.01 | - |

### 5.3.2 Real-world data

**ModelNet10**. modelnet10 is a subset of the larger ModelNet40 [53] corpus, containing geometrically rather *simple* objects with few characteristic topological features. Considering the latter, unsurprisingly, all neural network baselines perform much better (by more than 20 percentage points) than any purely PH-based approach, regardless of the simplicial complex. Nonetheless, we observe that all PH-based approaches yield accuracies notably higher than the ≈54% reported in [17] (on the same benchmark data) for a method that *predicts* vectorized persistence barcodes via a neural network.

**Mechanical Components Benchmark (MCB)**. As a slightly more challenging dataset, we selected a (11-class) *subset* of meshes, dubbed mcb-c, from the publicly available MCB corpus [32], focusing on objects with more geometrically and topologically interesting features, see Appendix A.4. Comparing the neural network results of Table 3 with the results in [32, Table 5] (mostly >90% on the *full* dataset of 68 classes), we see that the increased object complexity (of our 11-class subset) manifests as a drop in classification accuracy. Notably, Flood PH is competitive with the neural networks and achieves a significantly better balanced accuracy than Alpha$^\dagger$ PH and Alpha$^\dagger$ PH (avg.)

**Coral mesh dataset**. Finally, we curated a challenging dataset of 3D point clouds by uniformly sampling 1M points from surface meshes of *corals* from the *Smithsonian 3D Digitization* initiative. Our results on the *binary* classification problem of distinguishing corals by genus show that Flood PH yields (by far) the best result, suggesting that capturing fine details in the surface structure is mandatory for extracting discriminative information. Similar to Figure 5, we observed that the large number of points in the persistence diagrams of Alpha PH variants (compared to the leaner diagrams produced by Flood PH) tends to confound the persistence diagram vectorization technique, leading to drops in downstream performance. Moreover, all neural network baselines achieve only a balanced accuracy of less than 60% with a high variance across splits.

## 6 Discussion

We introduced the *Flood complex*, a novel simplicial complex designed to address the long-standing computational challenges of PH on large-scale point clouds. By combining careful subsampling and GPU parallelism, the Flood complex enables efficient computation of accurate PH approximation on point clouds with millions of points within seconds, offering speed increases of up to two orders of magnitude over the widely used Alpha complex. The Flood complex opens up several research questions for future work, including strengthening its theoretical guarantees and developing even more efficient algorithms. Specifically, we anticipate that the theoretical guarantees can be extended to Euclidean spaces of arbitrary dimension, and that more direct proofs, yielding tighter bounds on approximation quality and stability with respect to landmarks, are possible. Moreover, exploring additional real-world applications that require computing PH on large point clouds would further highlight the value of the Flood complex as a lightweight tool, and we envision that the Flood complex will facilitate a more widespread use of PH in machine learning applications that have, so far, been limited by computational constraints.

**Acknowledgments**

This work was supported by the Federal State of Salzburg within the EXDIGIT project 20204-WISS/263/6-6022 and projects 0102-F1901166- KZP, 20204-WISS/225/197-2019, the Austrian Science Fund (FWF) under project 10.55776/DFH4791124 (REVELATION). M. Uray and S. Huber were supported by the Christian Doppler Research Association (JRC ISIA), the Austrian Federal Ministry for Digital and Economic Affairs and the Federal State of Salzburg.

The authors also thank the anonymous reviewers for the valuable feedback during the review process, and Karsten Borgwardt and the Max Planck Institute of Biochemistry for providing access to the computing infrastructure.

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

# Supplementary Material

*In following supplementary parts to the main manuscript, we discuss implementation details & provide dataset descriptions (in Appendix A), some additional experiments (in Appendix B), and all proofs (in Appendix C) for our theoretical results.*

## A    Additional details

### A.1    Implementation details of masking and flooding procedures

In this section, we provide an algorithm for computing the *masking* and *(approximate) flooding process* tailored to modern GPU architectures.

First, recall that we compute for each simplex $\sigma \in \mathrm{Del}(L)$ its (approximate) filtration value $f(\sigma) = \max_{p \in P_\sigma} \min_{x \in X} d(p, x)$ with $P_\sigma \subset \mathrm{conv}(\sigma)$ a finite set of points on the convex hull of $\sigma$.

As described in Section 4, to avoid computing all $\sum_{\sigma \in \mathrm{Del}(L)} |P_\sigma| \cdot |X|$ pairwise distances $d(p, x)$, we identify for each simplex $\sigma$ a subset of points from $X$ over which the minimum $\min_{x \in X} d(p, x)$ is achieved. Specifically, this set is $B_{\sqrt{2}r_\sigma}(c_\sigma) \cap X$, where $B_{r_\sigma}(c_\sigma)$ is an enclosing ball of $\sigma$ computed using a variation of Ritter's heuristic. Checking for all simplices $\sigma \in \mathrm{Del}(L)$ which points $x \in X$ are within $B_{\sqrt{2}r_\sigma}(c_\sigma)$ requires $|\mathrm{Del}(X)| \cdot |X|$ distance evaluations. In fact, by presorting $X$ along one coordinate axis, we can compute, for each simplex $\sigma$, a slice $\tilde{X}_\sigma$ of $X$ enclosing the ball $B_{r_\sigma}(c_\sigma)$ via two binary searches, leading to an $O(\sum_\sigma |\tilde{X}_\sigma| + |X| \log |X| + |\mathrm{Del}(L)| \log |X|)$ complexity, which is usually much faster than doing all $|\mathrm{Del}(L)| \cdot |X|$ distance evaluations. The distances $d(c_\sigma, x)$ can then be computed independently, and are therefore straightforward to parallelize, yielding a Boolean masking matrix of shape $|\mathrm{Del}(L)| \times |X|$.

To be more precise, in our implementation, we also presort the simplex centers $c_\sigma$ along the same coordinate axis. This allows for efficient selection of a common bounding slice $\tilde{X}_B$ for an entire *batch $B$* of simplices to compute a masking matrix of shape $b \times \tilde{X}_B$, where $b = |B|$ is the batch size. The non-zero indices of the masking matrix are then, together with the points in $\tilde{X}_B$ and the sets $P_{\sigma_i}, i \in B$, input to a custom Triton [47] kernel, which efficiently computes the directed Hausdorff distances $f(\sigma_i)$.

We first initialize (with infinity) an array $A$ of shape $b \times |P_\sigma|$ in shared memory which collects the minimal distances $\min_{x \in X} d(p, x)$ for all $p \in P_B := (P_{\sigma_1}, \ldots, P_{\sigma_B})$. The kernel is then launched over a grid of multiple independent programs, each scheduled to run on the GPU's streaming multiprocessors. Each program receives the points in the slice $\tilde{X}_B$ and a chunk of simplex points $P_\sigma$ all from the same simplex $\sigma$. Moreover, it receives a chunk of non-zero $(\sigma, x)$ index pairs of the masking matrix, where we ensure that, for one chunk, all pairs correspond to the simplex $\sigma$. It then computes the minimal distances $\min_x d(p, x)$, where the minimum is taken over all $x$ in the chunk, and finally writes these values to $A$ via an atomic min operation. Once all programs terminate, the filtration value $f(\sigma_i)$ of each simplex in the batch is computed by taking the maximum of a row of $A$.

In fact, when representing each simplex $\sigma$ by a discrete set $P_\sigma$ defined in terms of barycentric coordinates as described in Section 4 (or any other discrete set $P_\sigma$ such that $P_\tau \subset P_\sigma$ for all faces $\tau \subset \sigma$), we can directly extract the filtration values of its faces by taking the maximum along the rows over appropriately selected column indices. It therefore suffices to perform the masking and distance computation only on batches of maximal simplices.

We additionally support a CPU implementation based on a $k$-d tree [5] data structure. Avoiding the masking step, we directly build a $k$-d tree on the points $X$ and compute the (global) minimal distance matrix $A$ (of shape $|X| \times |P_\sigma|$) using nearest-neighbor queries for each $p \in \bigcup_{\sigma \in \mathrm{Del}(L)} P_\sigma$.

## A.2 Source code

**Flooder source code.** We provide the full source code for constructing the Flood complex with subsequent PH computation at



https://github.com/plus-rkwitt/flooder



Furthermore, for easy use, practitioners can *install* (including all dependencies) our Python package flooder[2] (which is available on PyPi) via

```
pip install flooder
```

Below is a minimal working example (MWE) of how to compute Flood PH on 1M points sampled from the surface of a torus. Other examples, including timing experiments, can be found in the examples folder of the GitHub repository listed above.

```python
from flooder import (
    flood_complex, generate_landmarks, generate_noisy_torus_points_3d
)

DEVICE = "cuda"
n_pts = 1_000_000  # Number of points to sample from torus
n_lms = 1_000      # Number of landmarks for Flood complex

pts = generate_noisy_torus_points_3d(n_pts).to(DEVICE)
lms = generate_landmarks(pts, n_lms)

stree = flood_complex(pts, lms, return_simplex_tree=True)
stree.compute_persistence()
ph_diags = [stree.persistence_intervals_in_dimension(i) for i in range(3)]
```

**Datasets.** In addition to the Flood complex implementation, the flooder package provides all point cloud datasets used for the object classification experiments in Section 5.3. These datasets are ready-to-use, with all pre-processing steps already applied and come with pre-defined splits to ensure reproducibility. These topologically challenging datasets can serve as a standardized benchmark for future research in topological data analysis and machine learning on point clouds in general. Below is a minimal working example (MWE) of how to load a dataset.

```python
from flooder.datasets import (
    CoralDataset, MCBDataset, RocksDataset,
    ModelNet10Dataset, SwisscheeseDataset,
)

dataset = CoralDataset('./coral_dataset/')
for data in dataset:
    print(data.x.shape, data.y)
```

**Experiments code.** Experiments can be reproduced using the following repository:



https://github.com/plus-rkwitt/flooder-experiments



## A.3 Computing infrastructure

The main experiments were run on an SUSE Linux Enterprise Server 15 SP6 system with AMD EPYC 9554 64-Core Processors, 1024 GB of main memory, and NVIDIA H100 80GB HBM3 GPUs.

---

[2]all experiments were run with flooder (v1.0rc5)

### A.4 Dataset details

In the following, we provide a more detailed description of the used datasets and their properties.

**Corals.** We collected and curated a challenging 3D point cloud classification dataset by uniformly sampling 1M points from surface meshes of *corals* obtained from the *Smithsonian 3D Digitization* initiative[3]. In particular, this dataset is a *subset* of all coral meshes that are available under CC0 license, classified by *genus*. We label the corals according to their genus, and only use classes that have at least 30 instances. Overall, this yields a binary classification problem with a total of 83 available coral meshes. Specifically, there are 31 corals with the genus Acroporidae and 52 with Poritidae. On average, the meshes have ≈ 900k vertices (ranging from ≈ 29k to ≈ 10M with median ≈ 500k). A rendered example (with 1818450 vertices) is shown in Figure 6. In particular, the coral

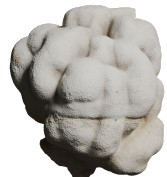

**Figure 6:** One example of a *Poritidae* coral.

*Leptoseris paschalensis* (USNM 53156), which is the mesh with the most vertices in the collection, is also used to showcase the capabilities of the Flood complex in computing PH on large point clouds; see Section 5.1.

**Mechanical Components Benchmark (MCB).** We used a *subset*, dubbed `mcb-c`, of the publicly available MCB-A dataset [32] to assess the classification performance on geometrically and topologically challenging objects. We filter MCB-A by (1) mesh size and (2) class size. In particular, we only take meshes with more than 10k vertices and then keep classes with more than 30 remaining instances. Here, the *vertex count* of a mesh serves as a proxy of *geometrical complexity*, with the intuition that object meshes with a larger number of vertices tend to be geometrically more complex. After filtering, 1,745 meshes split into 11 classes remain (out of 68 original classes), with ≈ 34.5k vertices on average. Finally, we uniformly sample 1M points from each mesh surface to obtain our training/testing point clouds.

**ModelNet10.** `modelnet10`[4] is a subset of the larger ModelNet40 benchmark [53] with 10 object categories (bathtub, toilet, table, etc.) distributed across 4,899 surface meshes. Originally, we have 3,591 training and 1,308 testing instances. We aggregate all meshes and then create ten random 80/20% training/testing splits. On average, we have ≈ 9.5k vertices per mesh. Overall, this dataset contains geometrically rather *simple* objects with few characteristic topological features. From each surface mesh we uniformly sample 250k points.

**Rhinovirus.** This point cloud is obtained from a cryo-electron-microscopy reconstruction [49] of the protein structure of RV-A89[5]. The raw (density) data is provided as voxels, which we first smooth out by applying a 3D average pooling (with kernel size 3, stride 1 and padding 1). We then transform it into a point cloud by taking the centers of the voxels that pass the provided density threshold (of 0.151). This results in $|X| \approx 12.3M$ points.

**Rocks.** We created a point cloud dataset mimicking porous materials using the `PoreSpy` library [23]. In particular, we create 1k Boolean voxel grids of size $256^3$, whose true values define the point clouds. In addition, we compute the *surface area* of a voxel grid as the sum over all voxels of the differences between the voxel grid and its shift by 1 along an axis. Following this strategy, we create 500 voxel grids using the *fractal noise* data generator and 500 using the *blobs* generator. In both cases, we select the porosity hyperparameter from a uniform distribution on [0.05, 0.95], resulting in point cloud sizes between 800k and 16M. The hyperparameters that affect the surface area (i.e., frequency for fractal noise and "blobiness" for blobs) are empirically tuned so that a wide range of (porosity, surface) tuples is approximately uniformly covered, see Figure 7. For details, we refer to the source code.

## B  Additional experiments

### B.1  Runtimes on different GPUs

We compare the runtime for computing Flood PH using different GPU architectures. Specifically, we report runtimes (in s) on an NVIDIA GeForce RTX 2080 Ti, a GeForce RTX 3090 and a H100 NVL

---

[3]available at https://3d.si.edu/corals

[4]available at https://modelnet.cs.princeton.edu

[5]available at https://www.emdataresource.org/EMD-50844

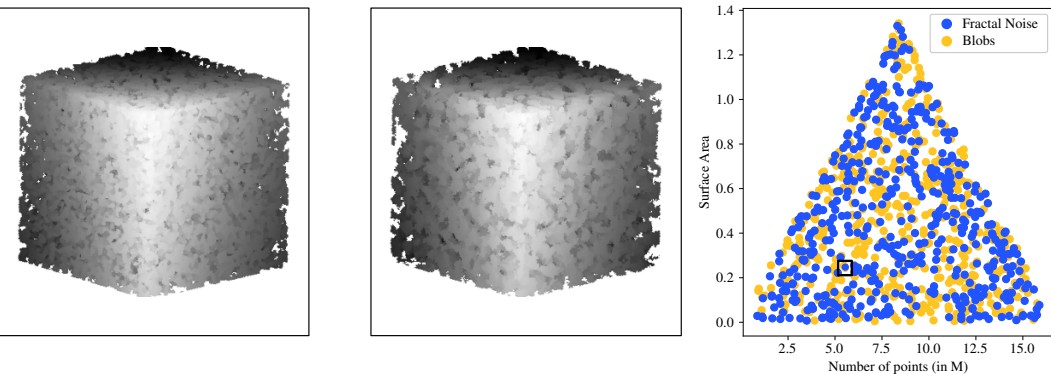

**Figure 7:** Two point clouds from the `rocks` dataset with similar porosity and surface area, one created using the blobs generator (**left**) and one using the fractal noise generator (**middle**). The **right** panel shows the number of points plotted against surface area (i.e., the regression target) for both generators. The black square □ indicates the location of the two examples.

card. As in Section 5.2, we evaluate on the *vertices* of the meshes used for generating the `corals` dataset and show results in Figure 8.

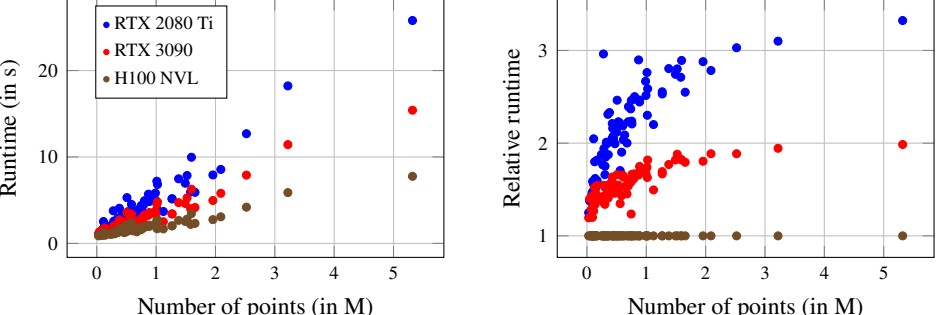

**Figure 8:** Absolute (**left**) and relative (**right**) runtimes of Flood complex construction plus persistent homology computation on the vertices of coral meshes on different GPUs. Hyperparameters were 2048 landmarks, 1024 random points and batch size 256.

### B.2 Additional results on large-scale point clouds

We present additional results to Section 5.1. Figure 9 extends Figure 2 and shows *all* persistence diagrams from the RV-A89 virus and the Leptoseris paschalensis coral. Figure 10 shows bottleneck distances to Alpha PH on the full coral point cloud for varying numbers of landmarks. When comparing to Figure 2, one notices the much larger runtimes of Flood PH on the coral point cloud, which is increased by a factor of around 2 when many landmarks are used, e.g., 27 versus 14 seconds when using 5k landmarks and 35 versus 17 seconds when using 10k landmarks. On closer inspection, this is caused by the masking step of Flood PH being more successful for the latter, resulting in only around half of the distance comparisons that need to be done for computing the filtration values.

### B.3 Effect of landmark selection method

In Table 4, we compare the bottleneck distances between Alpha PH on the full large-scale point clouds from Section 5.1 when landmarks are selected using either farthest point sampling (FPS) or uniform random sampling (abbr. as Rand). We report results obtained when using 1k landmarks and filtration values are computed from a grid with 30 points per edge. Specifically, we report mean and standard deviations from 25 runs, where the randomness in FPS comes from different initial points. As expected, FPS achieves a better coverage of the point clouds, resulting in lower Hausdorff distances between landmarks and the point cloud, and consequently lower bottleneck distances to the persistent homology of Alpha PH (full).

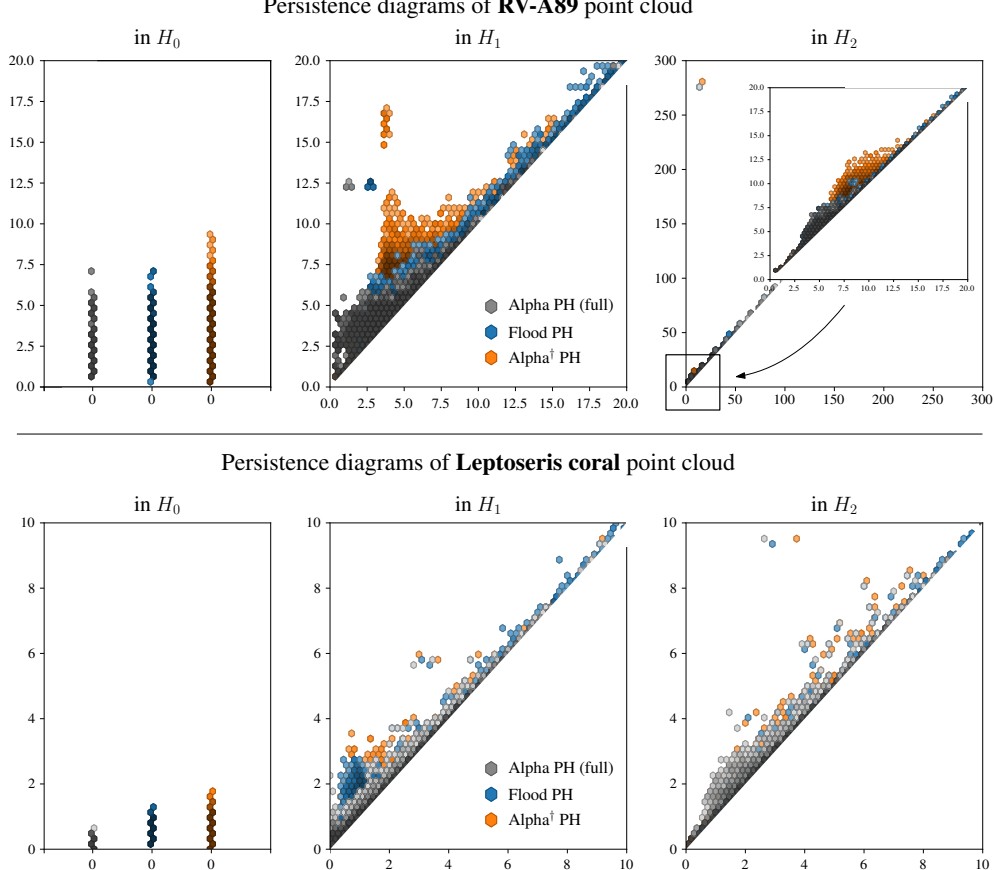

**Figure 9:** Hexbin plots of persistence diagrams of the RV-A89 and Leptoseris coral point clouds. Gray is Alpha PH on the full point cloud, blue is Flood PH with 10k landmarks, orange is Alpha PH on a subset of size 175k.

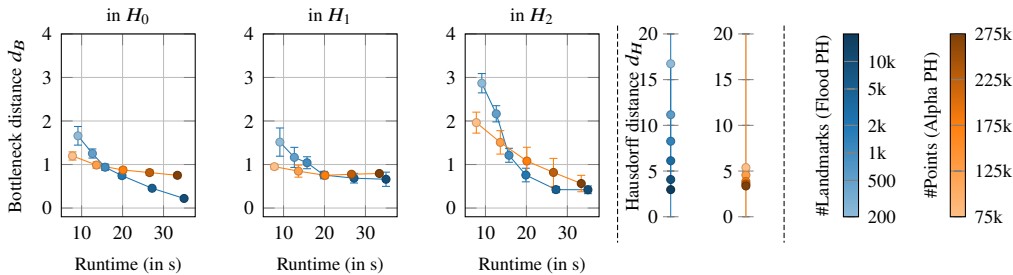

**Figure 10:** *Approximation quality* of Flood PH and Alpha PH on the Leptoseris paschalensis coral. The **(left)** panel shows bottleneck distances to Alpha PH (full) in $H_0$, $H_1$ and $H_2$ when varying the number of landmarks for Flood PH and the subsample size for Alpha PH. The **(middle)** panel shows the Hausdorff distance between the full point cloud and the landmarks, resp., the points in the subsample. The **(right)** panel shows the color coding used in all the plots.

**Table 4:** Comparison of bottleneck distances ($d_B$) between Alpha PH (full) and Flood PH when selecting landmarks using FPS and uniformly random (Rand) sampling on the large-scale point clouds from Section 5.1.

|  | RV-A89 | | Leptoseris Paschalensis | |
|---|---|---|---|---|
|  | FPS | Rand | FPS | Rand |
| Hausdorff distance $d_H(L, X)$ | $44.7 \pm 0.2$ | $79.1 \pm 4.0$ | $8.2 \pm 0.1$ | $20.0 \pm 3.3$ |
| Bottleneck distance $d_B$ in $H_0$ | $1.7 \pm 0.1$ | $3.2 \pm 0.1$ | $0.9 \pm 0.1$ | $0.9 \pm 0.1$ |
| Bottleneck distance $d_B$ in $H_1$ | $3.9 \pm 0.8$ | $3.3 \pm 0.5$ | $1.0 \pm 0.1$ | $1.2 \pm 0.2$ |
| Bottleneck distance $d_B$ in $H_2$ | $5.7 \pm 1.1$ | $5.4 \pm 1.6$ | $1.2 \pm 0.2$ | $2.6 \pm 3.2$ |

## C  Theory

### C.1  Proofs

For $p \in \mathbb{R}^d$ and a bounded set $X \subset \mathbb{R}^d$, we define $d(p, X) = \min_{x \in X} d(p, x)$. We recall the definition of the *directed Hausdorff distance*

$$\vec{d}_H(X, Y) := \max_{x \in X} \min_{y \in Y} d(x, y) = \max_{x \in X} d(x, Y) \tag{7}$$

for bounded sets $X, Y \subset \mathbb{R}^d$.

For completeness, we include a proof that the directed Hausdorff distance satisfies a triangle inequality.

**Lemma 8.** *Let $(X, d)$ be a metric space and let $A, B, C \subset X$ be compact. Then,*

$$\max_{a \in A} d(a, C) \leq \max_{a \in A} d(a, B) + \max_{b \in B} d(b, C) \ . \tag{8}$$

*Proof.* Fix $a_0 \in A$. Since $A, B, C$ are compact, there are $b_0 \in B$ such that $d(a_0, B) = d(a_0, b_0)$ and $c_0 \in C$ such that $d(b_0, C) = d(b_0, c_0)$. Hence,

$$d(a_0, C) = \inf_{c \in C} d(a_0, c) \leq d(a_0, c_0)$$
$$\leq d(a_0, b_0) + d(b_0, c_0) = d(a_0, B) + d(b_0, C)$$
$$\leq \max_{a \in A} d(a, B) + \max_{b \in B} d(b, C) \ .$$

Since $a_0 \in A$ was arbitrary, we conclude

$$\max_{a \in A} d(a, C) \leq \max_{a \in A} d(a, B) + \max_{b \in B} d(b, C) \ .$$

$\square$

### Proof of Theorem 2

*Proof.* Let $\sigma \in \mathrm{Del}(L)$ and denote by $f, f' : \mathrm{Del}(L) \to \mathbb{R}^+$ the filter functions that define $\mathrm{Flood}(X, L)$ and $\mathrm{Flood}(X', L)$, respectively. Observe, that $f$ is the directed Hausdorff distance between $\mathrm{conv}(\sigma)$ and $X$, which satisfies the following triangle inequality (see Lemma 8),

$$f(\sigma) = \max_{p \in \mathrm{conv}(\sigma)} d(p, X) \leq \max_{p \in \mathrm{conv}(\sigma)} d(p, X') + \max_{x' \in X'} d(x', X) = f'(\sigma) + \max_{x' \in X'} d(x', X) \ ,$$

and therefore $f(\sigma) - f'(\sigma) \leq \max_{x' \in X'} d(x', X)$. As, similarly, $f'(\sigma) - f(\sigma) \leq \max_{x \in X} d(x, X')$, we conclude

$$|f(\sigma) - f'(\sigma)| \leq \max \left( \max_{x \in X} d(x, X'), \max_{x' \in X'} d(x', X) \right) = d_H(X, X') \ . \tag{9}$$

The main stability theorem from [15] then implies Eq. (2) which concludes the proof. $\square$

### Proof of Corollary 4

*Proof.* Upon combination of Theorems 2 and 3, and Hausdorff stability of Alpha complexes, we immediately get $\forall i \in \mathbb{N}$:

$$d_B \left( \mathrm{dgm}_i(\mathrm{Alpha}(X)), \mathrm{dgm}_i(\mathrm{Flood}(X, L)) \right)$$
$$\leq d_B \left( \mathrm{dgm}_i \mathrm{Alpha}(X)), \mathrm{dgm}_i(\mathrm{Alpha}(L)) \right) + d_B \left( \mathrm{dgm}_i(\mathrm{Alpha}(L)), \mathrm{dgm}_i(\mathrm{Flood}(X, L)) \right)$$
$$\leq d_H(X, L) + d_H(X, L) = 2 d_H(X, L) \ .$$

$\square$

**Proof of Theorem 5**

*Proof.* To prove the theorem, we need to show that $\mathrm{dgm}_i(\mathrm{Flood}(X, L))$ and $\mathrm{dgm}_i(\mathrm{Flood}(X, L, P))$ are $\max_{\sigma \in \mathrm{Del}(L)} d_H(P_\sigma, \mathrm{conv}(\sigma))$ interleaved. As already mentioned, $\mathrm{Flood}_r(X, L) \subset \mathrm{Flood}_r(X, L, P)$. Consequently, it suffices to show that $\mathrm{Flood}_r(X, L, P) \subset \mathrm{Flood}_{r+\epsilon}(X, L)$ for any $r \geq 0$ and $\epsilon < \max_{\sigma \in \mathrm{Del}(L)} d_H(P_\sigma, \mathrm{conv}(\sigma))$.

To this end, we assume that $\sigma \in \mathrm{Flood}_r(X, L, P)$, i.e., $P_\sigma \subset \bigcup_{x \in X} B_r(x)$, and let $\epsilon < \max_{\sigma \in \mathrm{Del}(L)} d_H(P_\sigma, \mathrm{conv}(\sigma))$. By assumption, for each $q \in \mathrm{conv}(\sigma)$, there is a point $p \in P_\sigma$ such that $d(q, p) \leq \epsilon$. Moreover, since $\sigma \in \mathrm{Flood}_r(X, L, P)$, for each $p \in P_\sigma$ there exists a point $x_p \in X$ with $d(p, x_p) \leq r$. Hence,

$$d(q, X) \leq d(q, x_p) \leq d(q, p) + d(p, x_p) \leq r + \epsilon \ , \tag{10}$$

and thus $\mathrm{Flood}_r(X, L, P) \subset \mathrm{Flood}_{r+\epsilon}(X, L)$. Eq. (5) then follows from [15]. $\qquad\square$

**Proof of Lemma 6**

*Proof.* Let $\sigma = \{v_0, \dots, v_k\} \subset \mathbb{R}^d$ a $k$-simplex and let $P_\sigma = \{p = \sum_{i=0}^k \lambda_i v_i \in: \sum_{i=0}^k \lambda_i = 1, m\lambda_i \in \mathbb{N}\} \subset \mathrm{conv}(\sigma)$ be the grid points. For any $q := \sum_{i=0} \mu_i v_i \in \mathrm{conv}(\sigma)$, there exists a point $p := \sum_{i=0} \lambda_i v_i \in P_\sigma$ such that $|\mu_i - \lambda_i| \leq 1/m$ for every $i = 0, \dots, k$. Now, let $\delta_i := \mu_i - \lambda_i$ and observe that $\sum_{i=0}^k \delta_i = 0$. It follows that

$$
\begin{aligned}
2 \|p - q\|^2 = 2 \left\| \sum_{i=0}^k \delta_i v_i \right\|^2 &= 2 \sum_{i,j} \delta_i \delta_j \langle v_i, v_j \rangle \\
&= 2 \sum_{i,j} \delta_i \delta_j \langle v_i, v_j \rangle - \sum_j \delta_j \sum_i \delta_i \|v_i\|^2 - \sum_i \delta_i \sum_j \delta_j \|v_j\|^2 \\
&= - \sum_{i,j} \delta_i \delta_j \left( \|v_i\|^2 + \|v_j\|^2 - 2\langle v_i, v_j \rangle \right) \\
&= - \sum_{i,j} \delta_i \delta_j \|v_i - v_j\|^2 \ ,
\end{aligned}
$$

and therefore that

$$\|p - q\|^2 = - \sum_{i<j} \delta_i \delta_j \|v_i - v_j\|^2 \leq \frac{1}{m^2} \sum_{i<j} \|v_i - v_j\|^2 \ . \tag{11}$$

Since this holds for any $q \in \mathrm{conv}(\sigma)$, we conclude $d_H(P_{\mathrm{conv}(\sigma)}, \mathrm{conv}(\sigma)) \leq \frac{1}{m} \sqrt{\sum_{i<j} \|v_i - v_j\|^2}$. $\qquad\square$

A similar result holds with high probability for *randomly sampled points* as shown next.

**Lemma 9.** *Let $P_\sigma \subset \mathbb{R}^d$ be a set of independently drawn points from a uniform distribution on $\mathrm{conv}(\sigma)$ of size $\tilde{O}(\epsilon^{-d} \ln(1/\delta))$. Then, with probability of at least $1 - \delta$ (over the choice of $P_\sigma$), we have*

$$d_H(P_\sigma, \mathrm{conv}(\sigma)) \leq \epsilon \, \mathrm{diam}(\sigma) \ . \tag{12}$$

*Proof.* Let $\epsilon > 0$, and let $Q_r = \{Q_1, \dots, Q_m\}$ be a partition of $\sigma$ into sets of diameter at most $r := \epsilon \, \mathrm{diam}(\sigma)$, which can be obtained, e.g., by taking a covering of $\sigma$ with balls $B_j$ of radius $r/2$ and letting

$$Q_j = B_j \setminus \bigcup_{k=1}^{j-1} B_k \ .$$

Note that $m$ is smaller than the covering number of $\mathrm{conv}(\sigma)$ for radius $r/2$, and therefore $m \leq (4 \, \mathrm{diam}(\sigma)/r)^d = (4\epsilon)^d$. By a standard *balls in bins* argument [37], we obtain

$$\mathbb{P} \left[ \max_{p \in \sigma} d(p, P_\sigma) > \epsilon \, \mathrm{diam}(\sigma) \right] = \mathbb{P} \left[ \max_{p \in \sigma} d(p, P_\sigma) > r \right] \leq \mathbb{P}[\exists j : Q_j \cap P_\sigma = \emptyset] \leq m e^{-|P_\sigma|/m} \ .$$

Taking $|P_\sigma| \geq (4/\epsilon)^d \left( d \ln(4/\epsilon) + \ln(1/\delta) \right) \geq m \ln(m/\delta)$ yields $\mathbb{P}[\max_{x \in \sigma} d(x, P_\sigma) > \epsilon \mathrm{diam}(\sigma)] \leq \delta$ which establishes the claim. $\qquad\square$

**Proof of Lemma 7**

*Proof.* Let $\sigma = \{v_0, \ldots, v_k\}$ and let $p = \sum_{i=0}^{k} \lambda_i v_i \in \text{conv}(\sigma)$, i.e., $\sum_{i=0}^{k} \lambda_i = 1$ with $\lambda_i > 0$. Since $\sigma \subset X$, $z = \arg\min_{x \in X} d(p, x)$ implies $d(z, p) \leq d(p, v_i)$ for each $i \in \{1, \ldots, k\}$. It therefore suffices to minimize $d(p, x)$ over $x \in B_{\min_i d(p,v_i)} \cap X$.

Below, we will show that if $B_r(c)$ is an enclosing ball of $\sigma$, then for any $p \in \text{conv}(\sigma)$ it holds that $B_{\min_i d(p,v_i)}(p) \subset B_{\sqrt{2}r}(c)$, and therefore that $\max_{p \in \text{conv}(\sigma)} \min_{x \in X} d(p, x) = \max_{p \in \text{conv}(\sigma)} \min_{x \in X \cap B_{\sqrt{2}r}(c)} d(p, x)$.

First, consider an arbitrary point $c \in \mathbb{R}^d$. Observe that

$$\|v_i - c\|^2 = \|v_i - p\|^2 + \|p - c\|^2 + 2\langle v_i - p, p - c \rangle$$

for each $v_i \in \sigma$, and therefore

$$\sum_{i=0}^{k} \lambda_i \|v_i - c\|^2 = \sum_{i=0}^{k} \lambda_i \|p - c\|^2 + \sum_{i=0}^{k} \lambda_i \|v_i - p\|^2 + 2 \sum_{i=0}^{k} \langle \lambda_i(v_i - p), p - c \rangle = \|p - c\|^2 + \sum_{i=0}^{k} \lambda_i \|v_i - p\|^2 \ ,$$

where the last equality follows from $\sum_{i=0}^{k} \lambda_i = 1$ and $\sum_{i=0}^{k} \lambda_i v_i = p$. In particular, if $B_r(c)$ is an enclosing ball of $\sigma$ (and therefore $\text{conv}(\sigma)$), then

$$\|p - c\|^2 = \sum_{i=0}^{k} \lambda_i \|v_i - c\|^2 - \sum_{i=0}^{k} \lambda_i \|v_i - p\|^2 \leq r^2 - \min_{i=0,\ldots,k} \|v_i - p\|^2 \ .$$

Hence, if $z \in B_{\min_i d(p,v_i)}(p)$, i.e., $\|z - p\| \leq \min_{i=0,\ldots,k} \|p - v_i\|$, then

$$\|z - c\| \leq \min_i \|v_i - p\| + \|p - c\|$$

$$\leq \min_i \|v_i - p\| + \sqrt{r^2 - \min_i \|v_i - p\|^2}$$

$$\leq \max_{0 \leq s \leq r} s + \sqrt{r^2 - s^2}$$

$$\leq \sqrt{2}r \ .$$

$\square$

## C.2 Proof of Theorem 3

In this section, we cover the *homotopy equivalence* between $\text{Flood}_r(X, X)$ and the union of balls $\bigcup_{x \in X} B_r(x)$. Specifically, we show that $|\text{Flood}_r(X, X)|$ is homotopy equivalent to $|\text{Alpha}_r(X)|$, which, according to the nerve theorem [24, Corollary 4G.3], is homotopy equivalent to $\bigcup_{x \in X} B_r(x)$.

### C.2.1 Background tools

Let us first collect some tools and background information on Voronoi diagrams and Delaunay triangulations which, in arbitrary dimensions, express themselves as a combinatorial complex.

**Voronoi complex.** Consider $X \subset \mathbb{R}^d$ to be a finite point set. Then, a Voronoi complex tessellates $\mathbb{R}^d$ into closest-neighbor cells around the points of $X$. That is, we define by $\text{VC}(x, X) = \{p \in \mathbb{R}^d : d(p, x) \leq d(p, X)\}$ the *Voronoi cell* of $x$ w.r.t. $X$. Below are some facts regarding Voronoi cells:

- A Voronoi cell is a $d$-dimensional convex polyhedron, i.e., the intersection of half spaces.
- A Voronoi cell is unbounded if and only if $x$ is on the convex hull of $X$; in other words, the Voronoi cell is a convex polytope if $x$ is in the interior of the convex hull of $X$.
- As a convex polyhedron, the Voronoi cell itself possesses a combinatorial complex structure, i.e., its lower-dimensional faces are convex polyhedra again, and so are the non-empty intersections.

The *Voronoi diagram* $\text{Vor}(X)$ is usually defined geometrically as the union of the boundaries of the Voronoi cells. From a combinatorial perspective, it is more convenient to define the Voronoi diagram

Vor$(X)$ as the union of the Voronoi cells as complexes, and Vor$(X)$ is then itself a complex of convex polyhedra. To emphasize this, we speak of the Voronoi complex and refer to the components of the Voronoi complex, Vor$(X)$, as *Voronoi faces*.

Any $k$-dimensional Voronoi face $f$ from Vor$(X)$ is characterized by $d - k$ points $x_1, \ldots, x_{d-k}$ of $X$ as follows: any point $x \in f$ is equidistant to $x_1, \ldots, x_{d-k}$, $d(x, x_i) \leq d(x, X)$ and, if $x$ is from the relative interior relint$f$ of $f$, then we even have $d(x, x_i) < d(x, X \setminus \{x_1, \ldots, x_{d-k}\})$. We call $x_1, \ldots, x_{d-k}$ the *defining points* of $f$.

**Delaunay complex.** We can naturally dualize a $k$-dimensional Voronoi face $f$ by the simplex $\sigma$ formed by its defining points $x_1, \ldots, x_{d-k}$. If we do this for the entire Voronoi diagram Vor$(X)$, we obtain the Delaunay complex Del$(X)$. We distinguish between the combinatorial simplex $\sigma = \{x_1, \ldots, x_{d-k}\}$ and its geometric realization $|\sigma| = \text{conv}(\sigma)$. It will be convenient to denote $f$ as $|\sigma|^{\dagger}$.

Note that $|\sigma|^{\dagger}$ and $|\sigma|$ are orthogonal, however $|\sigma|^{\dagger}$ might not intersect $|\sigma|$. So, let us denote by aff $|\sigma|^{\dagger}$ the affine hull of $|\sigma|^{\dagger}$, i.e., the smallest affine subspace supported by $|\sigma|^{\dagger}$ (e.g., in case of $|\sigma|^{\dagger}$ being a line segment, aff $|\sigma|^{\dagger}$ would be an infinite line). Then, if dim $|\sigma| < d$, the point $|\sigma| \cap$ aff $|\sigma|^{\dagger}$ is the closest equidistant point to the defining points of $\sigma$. Also note that the distance function $d(\cdot, \sigma)$ has no further local minima on aff $|\sigma|^{\dagger}$ and, in fact, is convex on aff $|\sigma|^{\dagger}$ (the Euclidean distance from a given point is convex on any affine subspace). In particular, if aff $|\sigma|^{\dagger}$ is of dimension 1, then walking along this line away from $|\sigma| \cap$ aff $|\sigma|^{\dagger}$ *increases* the distance to $\sigma$, and walking towards $|\sigma| \cap$ aff $|\sigma|^{\dagger}$ *decreases* this distance.

**Simplicial collapses.** Given an abstract simplicial complex $\Sigma$, it is often possible to simplify its structure while preserving the homotopy type of its geometric realization $|\Sigma|$ using simplicial collapses. Specifically, if $(\tau, \sigma) \in \Sigma$ are a pair of simplices such that $\tau \subset \sigma$ is a face of $\sigma$, dim $\sigma = \dim \tau + 1$ and $\tau$ has no other cofaces, then the removal of the pair $(\tau, \sigma)$ from $\Sigma$ is called an elementary collapse. Similarly, if dim $\sigma > \dim \tau$ and all cofaces of $\nu \supset \tau$ of $\tau$ are faces $\nu \subset \sigma$ of $\sigma$, then (starting from the top) there is a sequence of elementary collapses removing all such simplices $\nu$, including $\tau$ and $\sigma$. Such a sequence is called a (simplicial) collapse, and a simplex $\tau$ satisfying the assumption is called a *free face* of $\Sigma$. Importantly, simplicial collapses induce a deformation retraction between the geometric realizations of the complexes before and after the collapse, and therefore preserves the homotopy type. Consider a filtered complex $\{\Sigma_r : r \geq 0\}$ that stays constant between filtration values $i < i + 1$ but changes at $r = i + 1$ when a simplex $\tau$ is added. If, at $r = i + 1$, a simplex $\tau$ is added, resulting in a change of the homotopy type between $\Sigma_i$ and $\Sigma_{i+1}$, then we call the addition of $\tau$ a *critical event*. If instead multiple simplices are added at $i + 1$ but their removal is a simplicial collapse from $\Sigma_{i+1}$ to $\Sigma_i$, then we call their addition a *regular event*.

**Alpha complexes.** The Alpha complex Alpha$_r(X)$ with radius $r$ on a set $X \in \mathbb{X}^d$ is defined as the nerve of $\{\text{VC}(x, X) \cap B_r(x) : x \in X\}$, i.e., $\sigma \subset X$ is a simplex of Alpha$_r(X)$ if and only if $\bigcap_{x \in \sigma}(\text{VC}(x, X) \cap B_r(x)) \neq \emptyset$. By the nerve theorem, $|\text{Alpha}_r(X)|$ is homotopy equivalent to $\bigcup_{x \in X} B_r(x)$. Moreover, as a subcomplex of the Delaunay triangulation Del$(X)$, it is naturally embedded in $\mathbb{R}^d$. An equivalent definition of Alpha complexes is given by the filtration function

$$f_{\text{Alpha}} : \text{Del}(X) \to \mathbb{R}, \quad \sigma \mapsto \inf\left\{r \geq 0 : \bigcap_{x \in \sigma}(\text{VC}(x, X) \cap B_r(x)) \neq \emptyset\right\} . \tag{13}$$

**Lemma 10.** *Let $X \subset \mathbb{R}^d$ be in general position. A simplex $\tau \in \text{Del}(X)$ is a free face of $\text{Alpha}_{f_{\text{Alpha}}(\tau)}$ if and only if $|\tau| \cap \text{relint}|\tau|^{\dagger} = \emptyset$.*

*Proof.* Assume that $|\tau| \cap \text{relint}|\tau|^{\dagger} = \emptyset$. Then, the point $q \in |\tau|^{\dagger}$ that is closest to $\tau$ is on the boundary $\partial|\tau|^{\dagger}$. Hence, $q$ is also closest to other points $x \in X \setminus \tau$, i.e., there are $\sigma \supset \tau$ with $q \in |\sigma|^{\dagger}$ and $f_{\text{Alpha}}(\sigma) = f_{\text{Alpha}}(\tau) = d(q, \tau)$. Sorting these cofaces in descending order by their degree, we get a sequence of elementary collapses which will eventually remove $\tau$. For details, we refer to standard textbooks such as [20].

Conversely, assume $\{q\} = |\tau| \cap \text{relint}|\tau|^{\dagger}$. Then $q$ is the closest equidistant point to $\tau$ and therefore $f_{\text{Alpha}}(\tau) = d(q, \tau)$. Moreover, since $q \in \text{relint}|\tau|^{\dagger}$, there is no $\sigma \supset \tau$ with $q \in |\sigma|^{\dagger}$, and hence, no coface of $\tau$ is added at filtration value $f_{\text{Alpha}}(\tau)$. Hence, $\tau$ is not a free face of $\text{Alpha}_{f_{\text{Alpha}}(\tau)}$. $\square$

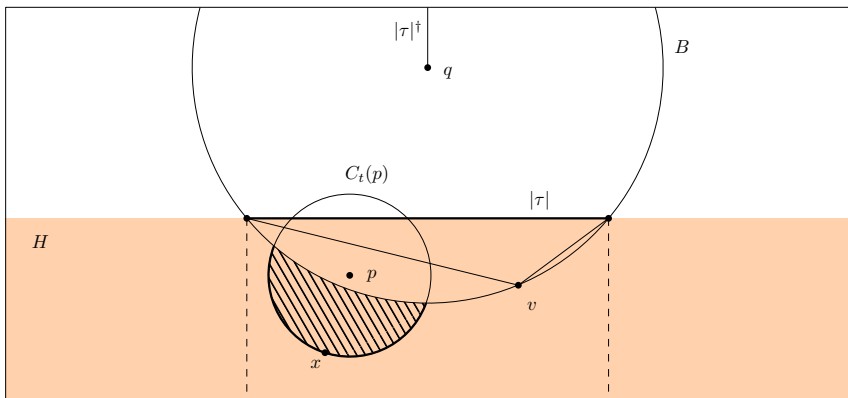

**Figure 11:** The existence of $p \in \mathrm{VC}(x, X) \cap (B \cap H)$ implies that $x$ is on the circular arc that bounds the tiled region, and therefore $x \in H$.

### C.2.2 Properties of the Flood filtration

Throughout this and the subsequent section, we will only consider Flood complexes whose landmark set $L$ is the entire point set $X$. In this situation, the filtration function is given by

$$f_{\mathrm{Flood}} : \mathrm{Del}(X) \to \mathbb{R}, \quad \sigma \mapsto \max\{d(p, X) : p \in |\sigma|\} \ . \tag{14}$$

It is worth mentioning that the point $p \in |\sigma|$ that realizes $f_{\mathrm{Flood}}(\sigma) = d(p, X)$ cannot be in the interior of a Voronoi region $\mathrm{VC}(x, X)$ of some point $x \in X$.

**Lemma 11.** *Let $X \subset \mathbb{R}^2$ be in general position. A simplex $\tau \in \mathrm{Del}(X)$ is a free face of $\mathrm{Flood}_{f_{\mathrm{Flood}(\tau)}}$ if and only if $|\tau| \cap \mathrm{relint}|\tau|^\dagger = \emptyset$.*

*Proof.* Let $\{q\} = |\tau| \cap \mathrm{aff}|\tau|^\dagger$. Since $q$ is the unique point on $|\tau|$ that is equidistant to $\tau$, it is the point of $|\tau|$ that is contained in $X_r$ the latest, i.e., only after the largest radius $r$. Consequently, $q \in |\tau|$ implies $f_{\mathrm{Flood}}(\tau) = d(q, \tau)$. Moreover, $q \in \mathrm{relint}|\tau|$ implies that every coface $\sigma \supset \tau$ contains a point $q' \in |\sigma| \cap |\tau|^\dagger$ with $d(q', \sigma) > d(q, \tau)$. Therefore, $f_{\mathrm{Flood}}(\sigma) > f_{\mathrm{Flood}}(\tau)$, and $\tau$ is not a free face.

Assume that $|\tau| \cap \mathrm{relint}|\tau|^\dagger = \emptyset$. Then, the point $q \in |\tau|^\dagger$ that is closest to $\tau$ is on the boundary $\partial|\tau|^\dagger$. Hence, $q$ is also closest to other points $x \in X \setminus \tau$, i.e., there are $\sigma \supset \tau$ with $q \in |\sigma|^\dagger$. In fact, since $X \in \mathbb{R}^2$, $|\tau|$ must be an edge and $|\sigma| = |\tau| \cup \{v\}$ must be a triangle with additional vertex $v$. To show that $\tau$ is a free face, we need to show that $\sigma$ has the same filtration value, i.e., that the point $o \in |\sigma|$ that realizes $f_{\mathrm{Flood}}(\sigma) = d(o, X)$ satisfies $o \in |\tau|$. To this end, denote by $H$ the closed half-space bounded by $\mathrm{aff}|\tau|$ that contains $|\sigma|$, and denote by $B := B_{d(q,\sigma)}(q)$ the ball whose boundary is the circumcircle of $|\sigma|$. Observe that $o$ must be on a Voronoi edge, and thus, we need to show that $d(\cdot, X)|_{|\sigma|}$ increases along the Voronoi edges towards $|\tau|$. The latter is true, if every 1-simplex $\gamma \in \mathrm{Del}(X)$ whose Voronoi edge $|\gamma|^\dagger$ intersects $|\sigma|$ satisfies (i) $\gamma \subset H$, and (ii) the point of $\mathrm{aff}|\gamma|^\dagger$ with minimal distance to $\gamma$ is outside the interior $\mathrm{relint}|\sigma|$ (because $d(\cdot, \gamma)$ is convex on $|\gamma|^\dagger$) .

We first show $\gamma \subset H$. In fact, we show the (slightly) more general result that for any $x \in X$, $\mathrm{VC}(x, X) \cap (B \cap H) \neq \emptyset$ implies $x \in H$. Obviously, the latter is true, if $x \in \tau$. Hence, from now on we assume $x \in X \setminus \tau$, as illustrated in Figure 11. By assumption, there is a point $p \in \mathrm{VC}(x, X) \cap (B \cap H)$. Since $p \in \mathrm{VC}(x, X)$, it holds that $t := d(p, x) < d(p, \tau)$. Further, $p \in B \cap H$, i.e., $p$ is in the minor circular segment of $B$ defined by $|\tau|$. In particular, $p$ is contained in the subset of $H$ that orthogonally projects onto $|\tau|$. Hence, the circle $C_t(p)$ with center $p$ and radius $t$ intersects the affine hull $\mathrm{aff}|\tau|$ only in a subset of the *segment* $|\tau|$. As $|\tau|$ is a chord of $B$, this implies that $C_t(p) \setminus B \subset H$. Since also $x \in C_t(p) \setminus B$, we conclude $x \in H$.

To finish the proof, we need to show that the point $c \in \mathrm{aff}|\gamma|^\dagger$ that minimizes the distance to $\gamma$ is not in the interior of $|\sigma|$. But this point $c$ is simply the midpoint of $|\gamma|$, and therefore $c \in |\gamma|$. Hence, $c \in \mathrm{relint}|\sigma|$ would imply that $|\gamma|$ intersects $\mathrm{relint}|\sigma|$, which contradicts that $|\mathrm{Del}(X)|$ is a well-defined geometric realization, i.e., that realizations of simplices only intersect in a common face. $\square$

**Lemma 12.** *Let $X \subset \mathbb{R}^d$ in general position and let $\sigma \in \mathrm{Del}(X)$. Then, $f_{\mathrm{Flood}}(\sigma) \leq f_{\mathrm{Alpha}}(\sigma)$ with equality if and only if $|\sigma| \cap |\sigma|^\dagger \neq \emptyset$.*

*Proof.* Denote by $c$ the center of the minimal enclosing ball of $\sigma$, observe that $c \in |\sigma|$, and that $f_{\mathrm{Flood}}(\sigma) \leq d(c, \sigma)$. Assume that $f_{\mathrm{Alpha}}(\sigma) = r$. Then, $|\sigma|^\dagger \cap \bigcap_{x \in \sigma} B_r(x) \neq \emptyset$, and therefore $c \in \bigcap_{x \in \sigma} B_r(x) \neq \emptyset$. In particular, $c \in \bigcap_{x \in \sigma} B_r(x)$, and thus $f_{\mathrm{Flood}}(\sigma) \leq d(c, \sigma) \leq r = f_{\mathrm{Alpha}}(\sigma)$. Herein, equality holds if and only if $|\sigma|^\dagger \cap \bigcap_{x \in \sigma} B_r(x) = \{c\}$. The lemma then follows because $c \in |\sigma|$ and $|\sigma| \cap |\sigma|^\dagger$ is either empty or a singleton $\{c\}$. $\qquad\square$

### C.2.3 Proof of the theorem

**Lemma 13.** *Let $X \subset \mathbb{R}^2$ be in general position and let $\sigma \in \mathrm{Del}(X)$. The addition of $\sigma$ to $\mathrm{Alpha}(X)$ is a critical event if and only if the addition of $\sigma$ to $\mathrm{Flood}(X)$ is a critical event. Moreover, if both are true, then, $f_{\mathrm{Flood}}(\sigma) = f_{\mathrm{Alpha}}(\sigma)$.*

*Proof.* Let $\sigma \in \mathrm{Del}(X)$. The addition of $\sigma$ to $\mathrm{Flood}(X, X)$ or $\mathrm{Alpha}(X)$ is a regular event if and only if $\sigma$ is a free face or a coface of a free face in $\mathrm{Flood}_{f_{\mathrm{Flood}}(\sigma)}$ or $\mathrm{Alpha}_{f_{\mathrm{Alpha}}(\sigma)}$, respectively. By definition, the addition of $\sigma$ is a critical event if it is not regular. The lemma then follows directly from Lemmas 10 to 12. $\qquad\square$

**Theorem 14.** *Let $X \subset \mathbb{R}^2$ be in general position. Then, $\mathrm{Flood}_r(X, X)$, $\mathrm{Alpha}_r(X)$ and $\bigcup_{x \in X} B_r(x)$ have the same homotopy type for any $r \geq 0$.*

*Proof.* The homotopy equivalence between $\mathrm{Alpha}_r(X)$ and $\bigcup_{x \in X} B_r(x)$ follows directly from the nerve theorem, see also [19] for the construction of a particular deformation retraction. Hence, it suffices to show the homotopy equivalence between $\mathrm{Flood}_r(X, X)$ and $\mathrm{Alpha}_r(X)$. To this end, we will construct a sequence of simplicial collapses from $\mathrm{Flood}_r(X, X)$ onto $\mathrm{Alpha}_r(X)$ for each $r$ where each collapse is an elementary collapse of an (edge, triangle) pair.

Let $0 = t_0 < t_1 < \cdots < t_\infty = \infty$ be the ordered list of filtration values for which a simplex $\sigma \in \mathrm{Del}(X)$ exists with $f_{\mathrm{Flood}}(\sigma) = t_i$ or $f_{\mathrm{Alpha}}(\sigma) = t_i$. We prove by induction on the $t_i$.

**Induction hypothesis.** For any $t_i$, there exists a sequence of simplicial collapses from $\mathrm{Flood}_r(X, X)$ to $\mathrm{Alpha}_r(X)$.

**Induction start.** At filtration value $t_0 = 0$, it holds that $\mathrm{Flood}_0(X, X) = X = \mathrm{Alpha}_0(X)$.

**Induction step.** Assume that there exists a sequence of collapses from $\mathrm{Flood}_{t_i}(X, X)$ onto $\mathrm{Alpha}_{t_i}(X)$. We will modify this sequence to get a sequence of collapses from $\mathrm{Flood}_{t_{i+1}}(X, X)$ onto $\mathrm{Alpha}_{t_{i+1}}(X)$. *For brevity, we will denote $F_t := \mathrm{Flood}_t(X, X)$ and $A_t := \mathrm{Alpha}_t(X)$.*

Given a filtration value $t_{i+1}$ there are three non-exclusive possibilities:

(i) There is $\sigma \in \mathrm{Del}(X)$ such that $f_{\mathrm{Flood}}(\sigma) = t_{i+1}$ but $f_{\mathrm{Alpha}}(\sigma) > t_{i+1}$.

(ii) There is $\sigma \in \mathrm{Del}(X)$ such that $f_{\mathrm{Alpha}}(\sigma) = t_{i+1}$ but $f_{\mathrm{Flood}}(\sigma) < t_{i+1}$.

(iii) There is $\sigma \in \mathrm{Del}(X)$ such that $f_{\mathrm{Flood}}(\sigma) = t_{i+1} = f_{\mathrm{Alpha}}(\sigma)$.

For now, assume that at each filtration value $t_{i+1}$ only one case is true.

Case (i). By Lemma 13, the addition of each such $\sigma$ to $\mathrm{Flood}_{t_i}(X, X)$ is a regular event. Specifically, $F_{t_{i+1}} = F_{t_i} \cup f_{\mathrm{Flood}}^{-1}(\{t_{i+1}\})$, and the removal of $f_{\mathrm{Flood}}^{-1}(\{t_{i+1}\})$ from $F_{t_{i+1}}$ is a simplicial collapse. Thus we can simply prepend this collapse to the existing sequence of collapses given by the induction hypothesis.

Case (ii). Similarly as above, Lemma 13 implies that the addition of $\sigma$ to $A_{t_i}$ is regular event and does not change its homotopy type. Specifically, $A_{t_{i+1}} = A_{t_i} \cup f_{\mathrm{Alpha}}^{-1}(\{t_{i+1}\})$, and the removal of $f_{\mathrm{Alpha}}^{-1}(\{t_{i+1}\})$ from $A_{t_{i+1}}$ is a simplicial collapse. Since $F_r \subset A_r$ for any $r \geq 0$, the pair $f_{\mathrm{Alpha}}^{-1}(\{t_{i+1}\})$ was already added to $\mathrm{Flood}_r$ at a previous filtration time, thus (see (i)), its removal must be in our sequence of simplicial collapses from $F_{t_i}$ onto $A_{t_i}$. Thus, the obvious candidate for a sequence from

$F_{t_{i+1}}$ onto $A_{t_{i+1}}$ is to take the existing sequence of collapses but to omit collapsing $f_{\text{Alpha}}^{-1}(\{t_{i+1}\})$. Note that collapsing $f_{\text{Alpha}}^{-1}(\{t_{i+1}\})$ can only create free faces that are in $A_{t_i}$, so the collapses in our sequence from $F_{t_i}$ onto $A_{t_i}$ are not affected by skipping the former. Thus, we have constructed a well-defined sequence of collapses from $F_{t_{i+1}}$ onto $A_{t_{i+1}}$.

Case (iii). The last case is $F_{t_{i+1}} \setminus F_{t_i} = A_{t_{i+1}} \setminus A_{t_i} =: \{\sigma_1 \ldots, \sigma_m\}$. The candidate sequence of simplicial collapses from $F_{t_{i+1}}$ onto $A_{t_{i+1}}$, is the sequence $F_{t_i}$ onto $A_{t_i}$ given by the induction hypothesis but applied to $F_{t_{i+1}}$. This is well defined, if any free face of $F_{t_i}$ whose removal is in the existing sequence of collapses is still free after the addition of $f_{\text{Flood}}^{-1}(\{t_{i+1}\})$ to $F_{t_i}$. However, $\tau \in F_{t_i}$ can only be such a free face if $\tau \notin A_{t_i}$, so by the assumption that $F_{t_{i+1}} \setminus F_{t_i} = A_{t_{i+1}} \setminus A_{t_i}$ no coface of $\tau$ is added to $F_{t_i}$ at filtration value $t_{i+1}$.

To finish the proof, we need to discuss the case when the filtration values are non-unique. By slightly changing the filtration values of the regular events, we get two new filtered complexes $\text{Flood}', \text{Alpha}' \subset \text{Del}(X)$ with homotopy equivalences $\text{Flood}'_r(X, X) \simeq \text{Flood}_r$ and $\text{Alpha}'_r \simeq \text{Alpha}_r$ for all $r \geq 0$ that satisfy the previously made assumption that cases (i) – (iii) are exclusive. □

### C.3 Example: the Flood complex of a circle

Herein, we will calculate $\text{Flood}(X, L)$ when $X := \{x \in \mathbb{R}^2 : \|x\| = 1\}$, i.e., the entire unit circle, and $L$ is selected using farthest point sampling (FPS). We will see that the deviation from the (persistent homology of) the thickenings $X_r$ depends only on *curvature* via the sagittas of circular arcs. In contrast, for an Alpha complex computed on $L$, the deviation depends on the *slope* via their chord lengths. This geometric difference manifests in the different decay of the approximation error in bottleneck distances: *linear* in the number of $L$ for the Alpha complex, but *quadratic* for the Flood complex, enabled by the additional information it can extract from $X$.

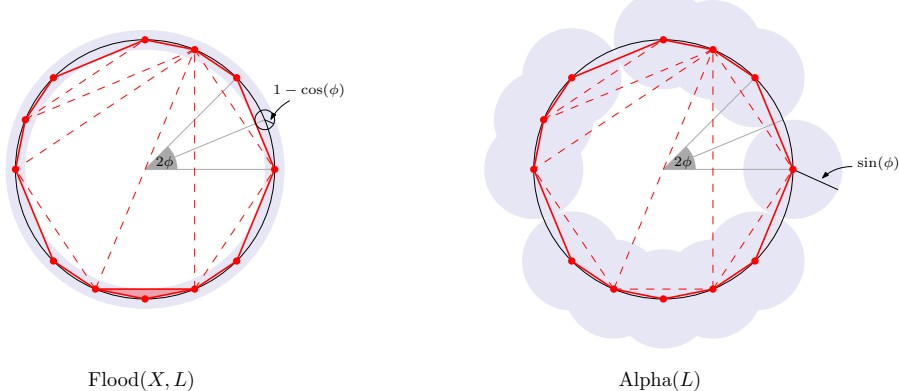

Flood$(X, L)$         Alpha$(L)$

**Figure 12:** Flood$(X, L)$ (**left**) and Alpha$(L)$ (**right**) for $X$ the unit circle and $|L| = 12$ landmarks (•) selected using FPS. The Flood complex matches the underlying topology already at lower filtration values.

**Landmark selection.** Without loss of generality, we assume that the initial point for FPS is at $(1, 0)$ corresponding to the angular coordinate of 0. FPS then proceeds in stages, where in the $k$-th stage all points with angular coordinate $2\pi/2^k$ are selected (in arbitrary order), if they have not been selected at an earlier stage. Once all $|L|$ landmarks are selected, the resulting Delaunay triangulation on $L$ is not unique. More specifically, it always contains $|L|$ edges along the circle connecting each landmark to the landmark with next lower and next higher angular coordinates, but the remaining $|L| - 3$ edges (across the circle) are arbitrary.

**Structure of persistence diagrams.** The persistent homology of Flood$(X, L)$ does not depend on the choice of Delaunay triangulation. In fact, the lifetimes in $H_0$ are just the lengths of the edges along the circle, which are always in Del$(L)$. Similarly, the only feature in $H_1$ is born as soon as all these edges are included in Flood$_r(X, L)$, and dies as soon as the entire convex hull of $L$ is flooded, which always occurs at filtration value 1.

**Filtration values of edges.** We will compute the filtration values of the edges that are relevant for persistent homology, i.e., of those along the circle. Their filtration values are just the distance from an

edge's midpoint to the circle $X$, i.e., the sagitta of the corresponding arc. Denoting the arc length by $\varphi$, this is just $1 - \cos(\varphi/2)$. Finally, we observe that there are exactly $m := 2(|L| - 2^{\lfloor \log_2(|L|) \rfloor})$ short arcs of length $\phi := \pi/2^{\lfloor \log_2(|L|) \rfloor}$ and $|L| - m$ long arcs of length $2\phi$. This is because if $|L|$ is a power of 2, then all $|L|$ arcs have length $2\pi/L$, and adding any further landmark splits one long arc and into two short ones, see Figure 12. We conclude that

$$
\text{dgm}_0(\text{Flood}_r(X, L)) = \left\{\left\{ \begin{array}{ll} (0, 1 - \cos(\phi/2)) & m\text{-times} \\ (0, 1 - \cos(\phi)) & (|L| - m)\text{-times} \\ (0, \infty) & \text{once} \end{array} \right\}\right\} , \tag{15}
$$

$$
\text{dgm}_1(\text{Flood}_r(X, L)) = \{\{(1 - \cos(\phi), 1)\}\} . \tag{16}
$$

**Bottleneck distances.** We compare the persistent homology of the Flood complex with that of the thickenings $\{X_r\}_{r \in [0,\infty)}$, which (by a slight abuse of notation) we denote as $\text{dgm}_i(X)$. It is straightforward to see that

$$
\text{dgm}_0(X) = \{\{(0, \infty)\}\} \text{ , and} \tag{17}
$$

$$
\text{dgm}_1(X) = \{\{(0, 1)\}\} . \tag{18}
$$

Hence, bottleneck distances are just

$$
d_B\big(\text{dgm}_0(X), \text{dgm}_0(\text{Flood}_r(X, L))\big) = \frac{1}{2}(1 - \cos(\phi)) \qquad \text{(by matching to the diagonal)} , \tag{19}
$$

$$
d_B\big(\text{dgm}_1(X), \text{dgm}_1(\text{Flood}_r(X, L))\big) = (1 - \cos(\phi)) . \tag{20}
$$

In particular, since $\phi < 2\pi/|L|$, we conclude that all bottleneck distances are smaller $\epsilon$ if $2\pi/|L| < \arccos(1 - \epsilon)$, i.e., if we use $|L| > 2\pi/\arccos(1 - \epsilon) \sim 2\pi/\sqrt{2\epsilon}$ landmarks.

**Comparison with Alpha complex.** We discard the additional information available through $X$ and simply compute an Alpha complex on $L$. Specifically, the arguments regarding the structure of persistence diagrams still apply, but now filtration values are not the sagitta of an arc, but half the chord length. Hence,

$$
d_B\big(\text{dgm}_0(X), \text{dgm}_0(\text{Alpha}_r(L))\big) = \frac{1}{2}\sin(\phi) , \tag{21}
$$

$$
d_B\big(\text{dgm}_1(X), \text{dgm}_1(\text{Alpha}_r(L))\big) = \sin(\phi) , \tag{22}
$$

and $|L| > 2\pi/\arcsin(\epsilon) \sim 2\pi/\epsilon$ are necessary.

# D Changelog

## D.1 Camera ready revisions / changes from arXiv version v1 to v2

We make the point cloud datasets used for the object classification experiments in Section 5.3 available as part of the `flooder` package. These datasets differ from the previous versions only (except for `mcb-c` and `modelnet10`, see below) in that they use different seeds for sampling the point clouds and also different training/validation/test splits. For reproducibility, we reran all experiments on the now public datasets and updated Tables 2 and 3 accordingly. On the `swisscheese`, `rocks` and `corals` datasets, the new results are (up to statistical uncertainty) in line with our initial results.

In the previous version of the manuscript, we selected a subset of `mcb` by joining MCB-A and MCB-B. This introduced duplicate point clouds (up to scaling and translation) in the dataset, however, with distinct labels. We now use only a subset of MCB-A. For `modelnet10`, we reduced the point clouds from 1M to 250k points to reduce the file size and facilitate sharing the data, and normalized point clouds to have coordinates in $[-1, 1]$. Moreover, we used different hardware for training the LGBM classification model, which uses the given training-time budget of 10 minutes more effectively. As a result, on `mcb-c` and `modelnet10`, balanced accuracies of the PH methods improved significantly while the neural network baselines are still consistent.

