# OpenReview forum: "The Flood Complex: Large-Scale Persistent Homology on Millions of Points"
_NeurIPS.cc/2025/Conference — NeurIPS 2025 poster_

### Official Review · Reviewer_b9wN · 2025-06-25

**Clarity:** 4
**Significance:** 2
**Originality:** 3
**Rating:** 5
**Confidence:** 4

**Summary:**

This paper proposes a novel filtration for point clouds the *Flood Complex*,
that approximates the well known Alpha complex, by using a smaller point cloud,
e.g., a subsample, while still preserving geometric information from the
original point cloud.

The paper provides theoretical approximation guarantees, as well as details to
compute an approximation and relaxation for a GPU implementation, which further
motivates this construction.
It finishes with an experimental section, showcasing the efficiency of the
algorithm as well as the statistical performances of the homology of this
filtration, which shows the practical the informativeness of this approach.

**Questions:**

- Can you comment on why the Flood complex performs better than the (more
 geometrically accurate) Alpha complex (when Alpha is computable) ?
 - Can you comment on the stability of the approach w.r.t. $L$ ?
 - Up to some proper quotient / filtrations, I feel that there is  a link between the
 perisistence image (c.f., [Persistent Homology for Kernels, Images, and
 Cokernels, 2009]) of the inclusion map of the Delaunay complex on $L$ into the
 alpha complex of $X$ and the persistent homology of your filtration.
 I wasn't able to formalize this, but I'm interested if you have the answer of
 this. This would also allow to define the Flood complex PH on other
 filtrations, e.g., Rips.
 - In Lemma 6. Isn't there a better (deterministic / entrop maximizing) way of
 putting the points in a simplex to better optimize this bound ? A similar
 statement can be said for (a,b)-standard measures [cuevas, 2004], but that's
 not very important.

**Ethical Concerns:**

["NO or VERY MINOR ethics concerns only"]

**Final Justification:**

All of my concerns were dealt with by the rebuttal and the following discussions. I'm keeping my score (accept).

**Limitations:**

yes.

**Paper Formatting Concerns:**

None in the main body.

There are some minor typos in the appendix, e.g., l895 or l927.

**Quality:**

4

**Strengths And Weaknesses:**

## Strengths
 - This filtration is intuitively reasonable.
 - The approximations guarantees are a must-have.
 - The relaxation to a GPU-friendly approach is very positive. This kind of
 optimization making TDA practical is very important for general applications
 in modern machine learning.
 - The proofs are sound.
 - This paper is very well written. Figure 1 gives a good intuition.
 - Experiments. The gain w.r.t. the full Alpha complex is very motivating.


## Weaknesses
 - Choice of $L$. Although the paper treat it by refering to fpsample, there
 are no guarantees given w.r.t this choice. There are some existing work on
 Witness complex that recovers some simplices of the delaunay complex for
 instance.
 - Choice of $L$ 2. The accuracies seem to reflect that this choice is not too
 impactful, but the paper do not provide an analysis on this choice. It would
 be interesting to see what (theoretically, and in practice) happens when we
 choose $L$ as a random (possibly non-uniform) subsample of $X$, and under some
 reasonable assumptions on the sampling measure of $X$. Such rates of
 convergence would allow to understand the tradeoff that we make when choosing
 the size of $L$.
 - Code. I installed the code and had issues with fpsample, so I couldn't run
 it. I manually checked the code, and it looked simple enough so that I could
 quickly read it. This is not a terrible issue, as this approach seems very
 easily implementable.
 - Code2. Unless I missed something, the code for the experiments is not given.
 - This is not fully on the GPU since a Delaunay complex is still needed in the
 end. This can be problematic in higher dimensions or if $L$ is too large (but
 is still better than the full Alpha complex).
 - It seems to me that this filtration is differentiable w.r.t. the initial
 point cloud, but this is not the case for its approximation on GPU.
 - Experiments. I was expecting to see bootstrap methods (e.g., Expected
 persistence diagrams / landscapes) or witness / sparse filtrations to be in
 the main body of the paper are they are natural competitors.
 - Experiments 2. I feel that they are too few datasets in the experiments.
 For instance, the only time the alpha complex could run was on an experiment
 in which it looses against the (less geometric informative) Flood complex.
 - Scope. I feel that the main contribution of this paper is in the field of
 computational geometry. I don't know how pertinent this work is to the NeurIPS
 audience.
 - Theorem 3. It would be very nice if this was true for any Euclidean space.

---

> ### Author Rebuttal · Authors · 2025-07-30
>
> *We thank the reviewer for the very positive feedback and the time spent. We address all weaknesses (W) and questions (Q) point by point, enumerated in the order they appeared in the review.*
>
> > **W1** Choice of $L$. Although the paper treat it by refering to fpsample, there are no guarantees...
>
> ▶ Thank you for pointing this out. We will remark that when using FPS, the Haussdorf distance between the landmarks $L$ and the point cloud $X$ decreases monotonically #landmarks increases. In fact, the landmarks, if selected this way, are an $\epsilon$-net of $X$, with $\epsilon$ the min. distance between the last selected landmark to any other landmark.
>
> > **W2** Choice of $L$. The accuracies seem to reflect that this choice is not too impactful, but the paper do not provide an analysis on this choice. It would be interesting to see what ...
>
> ▶ Our approx. guarantee (Cor. 4) wrt the full Alpha complex depends on the Hausdorff distance between $L$ and $X$. For an empirical study, we kindly refer to our response to **Sf3t**. In terms of theoretical aspects, we do not have a full answer yet. Still, if a sequence $(X_1,\dots,X_n)$ is drawn iid from a probability measure on a totally bounded subset of any metric space $(𝒳, d)$, then the minimal distance $d_n=\min_{i<n}d(X_i,X_n)$ satisfies $ℙ[d_n>\epsilon]\le𝒩(\epsilon/2)/(2n)$, where $𝒩$ denotes the covering radius of that subset. This result (and others) is proven in [Kulkarni et al. "Rates of convergence of nearest neighbor estimation under arbitrary sampling", In: IEEE Trans. on Inf. Theory '95]. In our setting, this means that if $X$ is an iid sample from a measure on a bounded subset of Euclidean space (and $L$ is considered as the first #landmarks points), then $ℙ[d_H(X,L)>\epsilon]\le 𝒩(\epsilon/2)/(2|L|)$, where $𝒩(r)$ scales as $r^{-d}$ where $d$ is the intrinsic dimension, i.e., $d=2$ for a surface (mesh) and $d=3$ for a volume. Stronger guarantees are possible if the point cloud geometry is taken into account.
>
> > **W3** ...issues with fpsample, so I couldn't run it...
>
> ▶ We are sorry to hear you had issues; `fpsample` worked on all of our test systems. If you still want to try out our code, you could use torch_cluster's FPS implementation as a replacement (or select $L$ randomly).
>
> > **W4** ...the code for the experiments is not given.
>
> ▶ You are correct, only a few examples were provided with our code. Unfortunately, we are not allowed to upload (or link to) additional material for the rebuttal. However, we will release the *full* code with all necessary parts for reproducibility after the reviewing phase.
>
> > **W5** This is not fully on the GPU since a Delaunay complex is still needed in the end. This can be problematic in higher...
>
> ▶ We agree. The Flood complex is designed for low-dim. point clouds as an efficient alternative to the Alpha complex. We want to add, however, that there is currently no viable option to compute large-scale PH in high dimensions. E.g., computing PH (up to dim. 8) of only 100 uniformly distributed points in the 8-dim. unit cube already takes ~1min when using an Alpha or a sparse Rips complex with sparsity parameter 1 (theoretical guarantees require <1) on our hardware.
>
> > **W6** It seems to me that this filtration is differentiable w.r.t. the initial point cloud, ... Would you be more specific in this point...
>
> ▶ The (approx.) filtration function is given by $f(σ)=\max_{p\in P_σ}\min_{x\in X}d(p,x)$, which is differentiable unless there are multiple minimizers (similar to max-pool. layers or L1 regularization). In practice, the Triton kernel prevents differentiability, as we did not implement the derivative function (*yet*). This will be done in the final release version.
>
> > **W7** Experiments. I was expecting to see bootstrap methods (...) or witness / sparse filtrations to be in the main body of the paper...
>
> ▶ You likely have seen it; we compare to averaged persistence vectorizations in our appendix (Table 2) and can move them to the main part. We also compare to sparse Rips in Fig. 2. Regarding witness complexes, these methods shine in higher dimensions (as does the Rips complex) but are neither competitive to the Alpha nor to the Flood complex in the low-dim. regime. To the best of our knowledge, there are no sparse versions of the Alpha complex.
>
> > **W8** Experiments 2. I feel that they are too few datasets...
>
> ▶ For our comparison to the Alpha complex, we consider two settings: one (Alpha-full) where the available information is the same, i.e., Alpha on $X$, and one (Alpha-†) where the available *runtime budget* is equal. However, in case of the former, runtime of the Alpha complex is prohibitive. Still, for the rebuttal we ran classification experiments using Alpha-full on `swisscheese` (93% ± 2%, 70s per point cloud) and `MCB` (60% ± 3%, 200s). Using more points does not improve accuracy, likely because of oversmoothing effects during vectorization. Moreover, the latter becomes slow and difficult to tune for large point clouds, e.g., vectorizing all Alpha-full diagrams from `MCB` takes 30min per split.
>
> To address your datasets concern, we generated a 2nd synthetic dataset using the `porespy` library [Gostick J et al. "PoreSpy: A Python Toolkit for Quantitative Analysis of Porous Media Images". JOSS, 2019. ]. Point clouds in this dataset (abbr. as `rocks`) are extracted from boolean voxel grids of shape (256,256,256) that were constructed from *two* different generators (blobs and fractal noise), yielding point clouds of varying size up to 16M points. We evaluate (1) classification accuracy wrt the data generating method (i.e., 2 classes) and (2) regression accuracy wrt the surface area computed from the voxel grid. Please see our results in response to **Sf3t**.
>
> > **W9** Scope. I feel that the main contribution of this paper is in the field of computational geometry. I don't know how pertinent this work is to the NeurIPS audience.
>
> ▶ We understand your point, but emphasize that our approach represents a critical advancement for the ML community, especially in domains where understanding data topology at scale is crucial. From our perspective, Flood PH facilitates TDA at modern data scales and thereby offers a way to integrate *global geometric reasoning* (via PH) into ML pipelines, which has been largely underutilized  due to computational constraints. Further, from a practical perspective, our work aligns well with what is prominently mentioned in a recent ICML position paper [Papamarkou et al. "Position: Topological Deep Learning is the New Frontier for Relational Learning". In: ICML 2024] in terms of availability of *efficient* TDA methods and software for ML problems.
>
> > **W10** Theorem 3. It would be very nice if this was true for any Euclidean space
>
> ▶ Yes, this would be great. So far, we only have a proof in 2D where things can easily be illustrated, but we conjecture the theorem to hold in higher dimensions as well. Most of the building blocks for the general result are already there. The only extension missing is of Lemma 10, i.e., that *"a simplex $𝜏\in\mathrm{Del}(X)$ is a free face of $\mathrm{Flood}_{f\_{\mathrm{Flood}(𝜏)}}$ iff $|𝜏|\cap\mathrm{int}|𝜏|^\dagger=\emptyset$."*
>
> ▶ **Q1** When Alpha is computable (which it is given enough time budget), one reason we identified is the inability of existing *vectorization strategies* to deal with massive persistence diagrams. Essentially, we observed over-smoothing effects (e.g., due to a large number of summations in the method we selected, but this similarly holds for other strategies) that confound the vectorization quality and thus negatively impact downstream classification performance. The rather small persistence diagrams of Flood PH are advantageous in this regard.
>
> ▶ **Q2** In 2D, the Flood complex is stable wrt $L$, which follows directly from Col 4. Admittedly, the multiplicative constant of 6 is quite large and a more direct proof (also in higher dimensions) would be more satisfying. The difficulty when studying stability wrt $L$ is that the resulting filtrations are defined on two different simplicial complexes, i.e., on $\mathrm{Del}(L)$ and $\mathrm{Del}(L')$. We tried using a common subdivision, but were unable to link the difference in filtration values to the Hausdorff distance between the landmark sets.
> In the special case that there is a bijection $L'\ni l_i'=l_i+\epsilon_i$ such that $\{l_i,\dots,l_k\}\in\mathrm{Del}(L)$ iff
> $\{l'_i,\dots,l'_k\}\in\mathrm{Del}(L')$ it can be shown that $\mathrm{Flood}_r(X,L)\subset\mathrm{Flood}\_{r+\epsilon}(X,L')$
>  and vice versa, and therefore $d_B(\mathrm{dgm}(\mathrm{Alpha}(X)),\mathrm{dgm}(\mathrm{Flood}(X,L))<\epsilon$ with $\epsilon=\max_i\epsilon_i\le d_H(L,L')$.
>
> ▶ **Q3** Extending the Flood filtration to other complexes such as Rips would be great. Unfortunately however, there is no natural inclusion $\mathrm{Del}(L) \hookrightarrow \mathrm{Alpha}(X)_r \subset\mathrm{Del}(X)$, unless no point of $X$ is inside a circumsphere of a simplex in $\mathrm{Del}(L)$, which can be considered an unlikely boundary case. (On the contrary, we would like $L$ to sample $X$ "well".)
>
> ▶ **Q4** You are correct regarding the uniform sampling. However, for efficiency reasons, it is not viable to optimize the point selection for each simplex separately based on its geometry. However, it is possible to select an evenly-spaced grid on the standard 3-simplex with $m$ points per edge (obtained from $\{(i,j,k,l)\in ℕ^4\colon i+j+k+l=m\}$) and use that as the barycentric coordinates of the simplex points. This approach has the benefit that filtration values of the faces can directly be computed from the 3-simplices by taking the max. over a subset of simplex points. We already used this *deterministic* approach to isolate the effect of landmark selection for the sensitivity analysis wrt the landmarks (see response to **S3ft**) and will support it in the upcoming code release.

---

> > ### Comment · Reviewer_b9wN · 2025-08-04
> >
> > Thank you for your detailed answers. They cover most of my concerns.
> >
> > > Choice of L.
> >
> > Thank you for these details. The new experiments compairing
> > FPS with random are convincing enough to me.
> > I still think that adding a little background result on the (minimax)
> > convergence rate of $\mathbb E [d_H(X,L)]$ for random samples (from, e.g., the
> > reference you gave, or something in the vein of [convergence rates of PDs in
> > TDA]) would be interesting to have in the final version.
> > I am not familiar with the technicalities of `fpsample`, but if similar result
> > exist, they would also be useful. If no such result exist, an empirical
> > statistical study would also be convincing (e.g., empirical variance /
> > confidence intervals).
> >
> > > W9. Scope.
> >
> > I agree that such approach are necessary for advencements of TDA in ML.
> > I think this paper can morally fit in the TDA + ML scope, but I could
> > understand that one may have a different opinon on this. As pointed out by
> > **Reviewer Sf3t**, a little comparison (with a fixed computational budget) with
> > the latest point cloud classifiers could be very convincing.

---

> ### Author Response · Authors · 2025-08-04
>
> Thank you for your fast response. We are happy to hear that we could already address most points.
>
> > Choice of L.
>
> When using FPS, we can obtain similar asymptotic results. Indeed, the Hausdorff distance between $L$ and $X$ decreases monotonically with $|L|$, and with order $d_H(X,L)\in O(D\cdot |L|^{-1/d})$, where $D$ is the diameter of the point cloud and $d$ its intrinsic dimension, i.e., $2$ for a surface (mesh) and $3$ for a volume.
>
> To be more precise, [González, Clustering to minimize the maximum intercluster distance, Theorem 2.2] shows that FPS yields an $\epsilon$-net of the point cloud, meaning that any two landmarks are at distance at least $\epsilon$ (packing) and that $d_H(X,L)\leq\epsilon$ (covering), for some $\epsilon >0$. Moreover, we have that (e.g., [Feldmann & Filtser, Highway Dimension: a Metric View, Lemma 6 ]), given a $\epsilon$-packing $L$ of a set with diameter $D$, its size satisfies $|L|\leq 2^{d \log(2D/\epsilon)}$. We therefore have that $\epsilon\leq 2D\cdot k^{-1/d}$.
>
> In fact, Hausdorff distance is the criterion in FPS for selecting the next landmark, and therefore can in principle be used as a stopping criterion instead of pre-defining the number of landmarks $|L|$.
>
> We will add this discussion and the one for random sampling in the final version of the manuscript, following your suggestion.
>
> > Scope
>
> Thank you for acknowledging that our approach is "*necessary for advencements of TDA in ML*" and that it "*fits in the TDA + ML scope*". We hope that our work will foster the currently limited synergy between ML and topology, also in real-world applications.
>
> Please also note that we have added in our last reply to **Sf3t** some preliminary (given the limited time we had) results with two stronger neural baselines, PointMLP and PVT. In line with the original experiments, Flood PH succeeds on the structurally more challenging `cheese`, `rocks` and (real-world) `corals` datasets, while *all* tested neural architectures struggle.

---

> > ### Comment · Reviewer_b9wN · 2025-08-05
> >
> > Thank you for these details and the extra experiments. I have no further questions.

---

> ### Comment · Area_Chair_tagK · 2025-08-05
>
> Dear Reviewer,
>
> Many thanks for engaging with the authors in this discussion thread. Your contribution is making the review process of Neurips better.
>
> I just wonder if you could also follow up with the authors' latest response to your concerns? Many thanks in advance.
>
> Best,
>
> Ac

---

### Official Review · Reviewer_Sf3t · 2025-07-01

**Clarity:** 2
**Significance:** 2
**Originality:** 3
**Rating:** 5
**Confidence:** 4

**Summary:**

This paper proposes the Flood complex to efficiently compute Persistence homology on large-scale point cloud data. Flood complex on a set of landmarks L contains all such simplifies of the Delaney complex on L that are covered by balls of radius r originating from data points X. Theoretical results shows that flood complex is an approximation of Alpha complex on X (alpha(X)), contains less simplifies than alpha(X), and due to efficient GPU implementation of computing filtration values it results in scalable and faster alternatives to alpha(X).

**Questions:**

1. In Figure 2, why Flood(X, L) total time the same as the Construction time? Does that mean the PH computation time is negligible? The same question for Fig. 4(a). Could you provide a more detailed breakdown of the total time in a table?

2. Can the Flood complex be extended to non-Euclidean data, e.g., graphs?

3. What is the computational complexity of Flood filtration construction?

I am willing to adjust my score based on a satisfactory resolution of Weaknesses 1-4 with experiments.

**Ethical Concerns:**

["NO or VERY MINOR ethics concerns only"]

**Final Justification:**

All my concerns are resolved. Hence, I recommend acceptance.

**Limitations:**

yes

**Quality:**

2

**Strengths And Weaknesses:**

Strengths:
1. The Flood complex is efficient to compute and scales to large-scale point clouds.
2. The Flood complex results in better point cloud classification accuracy than the  alpha complex on the landmarks in a reasonable time.

Weaknesses:
1. **Limited consideration of Baselines:** Baselines are limited to alpha complex and sparse rips complex only. There several recent subsampling/landmark based PH approximation methods such as [1,2] that construct filtration for scalability, yet they have not been considered in the experiments.

[1] Adaptive approximation of persistent homology, Journal of Applied and Computational Topology, 2024

[2] Topological Data Analysis with $\epsilon$-net Induced Lazy Witness Complex, International Conference on Database and Expert Systems Applications, 2019

2. **Sensitivity analysis:** The authors do not study the sensitivity of Flood Complex with  respect to the choice of Landmarks.

3. **Tradeoff between \#landmarks and Approximation quality:** It is important to understand the tradeoff between the number of landmarks and PH approximation quality. Although the authors empirically studied it in Figure 4(d), the paper does not give a recipe for how many landmarks are enough for some user-specified approximation quality in terms of bottleneck distance. As a result, it is hard to determine how many landmarks someone should use when they deploy Flood complex for their own datasets.

4. **Experimental results:**

(a) The authors should report the tradeoff between #Landmarks and Acc. of Flood complex across various datasets in Table 1.

(b) In Table 1, the improvement in Acc. (Due to Flood Complex) is negligible compared to Alpha+.

(c) Similarly, in Figure 3, it seems the accuracy of alpha(X) with 100k landmarks is almost the same as Flood complex with 500k landmarks. Does that mean that if someone uses alpha(X) with more landmarks, one could obtain better results than Flood complex? Perhaps the author can extend the x-axis to 500k points to clarify this.

d) In Figure 4, explain why Alpha is slower than the Flood complex? Is it due to using Gudhi, which is a CPU-based implementation in contrast to the Flood complex, which is GPU-based? Or is it because the Alpha complex is computed on the entire point cloud rather than on landmarks? Also, why is the dimension 2 column in Fig. 4(b) missing?

5. **Better clarification on the role of GPU in Flood complex construction:** Referring to the supplementary material, I noticed the flood complex construction is not entirely based on GPU, the authors still need to (1) select landmarks on CPU and (2) construct alpha complex on the landmark on CPU. Hence, the authors should clarify the reason for the faster running time of the Flood complex and exactly what rolethe  GPU plays there.

---

> ### Author Rebuttal · Authors · 2025-07-30
>
> *We thank the reviewer for the constructive/positive feedback and the time spent. We address all weaknesses (W) and questions (Q) point by point enumerated as in the review and will adjust the paper accordingly.*
>
> > **W1** Limited consideration of Baselines:
>
> ▶ Thank you, we will acknowledge both references. In fact, [1] and [2] are conceptually closer to the Rips than to the Alpha complex. [1] presents a method to compute a distance matrix between equivalence classes of points on which a Rips complex is built. [2] proposes a method to select landmarks for a lazy witness complex. Due to the underlying simplicial complex (which is much larger than an Alpha complex) in both works, the PH computation takes very long even on a few points.
>
> To mitigate this issue, we use edge collapses as in [Boissonnat & Pritam. "Edge Collapse and Persistence of Flag Complexes", SoCG '20], but *still* the number of landmarks / equivalence classes that can be used is limited, and we face additional problems, briefly described below.
>
> In case of [1], the selection strategy requires building a Delaunay triangulation on the *entire* point cloud, i.e., the very step we try to avoid with the Flood complex. To achieve a reasonable runtime, we first run FPS to obtain 2k points as input to their selection process. Balanced classification accuracies are ~50% on `swisscheese` and `corals`, and 45% on `MCB`.
>
> In case of [2], when constructing lazy witness complexes (in Gudhi), only computing filtration values already takes long. E.g., for 100 landmarks on 10k witnesses from the `swisscheese` data, complex construction takes ~20s., and for 1M witnesses ~20min.
> However, in the former setting, we again obtain only 50% acc. on `swisscheese`.
>
> > **W2** Missing sensitivity analysis wrt choice of Landmarks.
>
> We evaluate the sensitivity wrt the *landmarks selection* on the data from Sec. 5.1 in terms of the bottleneck distance $d_B$ to Alpha PH of the entire point cloud $X$. We compare FPS to (uniform) random selection (Rand). To isolate the effect of landmark selection, we compute the flooding radii based on a evenly spaced grid (instead of random, see response to **b9wn**) of points on each simplex. For reference, we also list results for **Alpha**† (on 75k points).
>
> Virus|FPS|Rand|Alpha†
> -|-|-|-
> $d_H(X,L)$|33.6|62.0 ± 2.3|26.2 ± 1.0
> $d_B-ℍ_0$|1.9|3.3 ± 0.1|3.6 ± 0.6
> $d_B-ℍ_1$|2.8|3.2 ± 0.4|7.3 ± 0.3
> $d_B-ℍ_2$|2.4|9.1 ± 2.1|7.9 ± 0.6
>
> Coral|FPS|Rand|Alpha†
> -|-|-|-
> $d_H(X,L)$|6.1|15.5 ± 3.2|5.3 ± 0.4
> $d_B-ℍ_0$|0.65|0.69 ± 0.07|1.16 ± 0.09
> $d_B-ℍ_1$|0.79|1.23 ± 0.06|0.93 ± 0.04
> $d_B-ℍ_2$|0.71|2.22 ± 0.38|2.01 ± 0.13
>
> Bottleneck distances (the metric of interest)  of Flood PH (FPS) are significantly smaller than for Flood PH (Rand). The same applies to Alpha†, notably despite Hausdorff distances  of Alpha† are *larger*.
>
> > **W3** Tradeoff between #landmarks and Approximation quality
>
> ▶ Approx. quality is upper bounded by the Hausdorff distance $d_H$ between $L$ and $X$, see Cor. 4. The latter decreases monotonically with #landmarks selected by FPS. For an empirical evaluation of #landmarks vs bottleneck distance in $ℍ_2$ on a `swisscheese` point cloud, we refer to Fig. 4 column (d). Further, we computed $d_B$ between diagrams of $\text{Flood}(X,L_{\text{FPS}})$ and $\text{Alpha}(X)$ on the two point clouds from Sec. 5.1. using $10^k$ Landmarks for $k\in \{2.75,2.875,...,3.5\}$.
>
> **Virus**
> Landmarks|562|749|1000|1333|1778|2371|3162
> -|-|-|-|-|-|-|-
> $d_H(X,L)$|56.7|50.3|44.8|39.8|35.7|31.6|28.3
> $d_B-ℍ_0$|1.8|1.7|1.9|1.9|1.6|1.7|1.3
> $d_B-ℍ_1$|6.9|5.1|3.6|3.8|2.7|2.7|2.7
> $d_B-ℍ_2$|7.6|5.5|4.3|4.2|4.2|4.2|2.9
>
> **Coral**
> Landmarks|562|749|1000|1333|1778|2371|3162
> -|-|-|-|-|-|-|-
> $d_H(X,L)$|10.7|9.3|8.3|7.3|6.5|5.7|5.0
> $d_B-ℍ_0$|1.2|1.0|1.0|0.8|0.8|0.7|0.6
> $d_B-ℍ_1$|1.0|1.1|0.9|0.7|0.8|0.8|0.6
> $d_B-ℍ_2$|2.1|1.7|1.3|1.0|0.6|0.4|0.5
>
> > **W4a** Tradeoff between #Landmarks and Acc. of Flood complex across various datasets in Table 1.
>
> ▶ Below, we list classification acc / mse as a function of the #landmarks. As **b9wN** requested another dataset, the table includes two *new* columns for `rocks`, i.e., one binary classification task (cls) and a surface area regression task (reg). For a description of the dataset, please see our comments to **b9wN**.
>
> #Landmarks|cheese|rocks (cls)|rocks (reg)|corals|MCB|Modelnet10
> -|-|-|-|-|-|-
> 20|0.81 ± 0.03|0.56 ± 0.07|0.068 ± 0.002|0.53 ± 0.12|0.51 ± 0.02|0.52 ± 0.02
> 100|0.95 ± 0.01|0.67 ± 0.04|0.021 ± 0.001|0.64 ± 0.10|0.58 ± 0.03|0.57 ± 0.02
> 500|0.98 ± 0.01|0.81 ± 0.03|0.0009 ± 0.0002|0.75 ± 0.11|0.61 ± 0.03|0.59 ± 0.01
> 1000|0.99 ± 0.01|0.84 ± 0.06|0.0006 ± 0.0001|0.76 ± 0.10|0.63 ± 0.03|0.59 ± 0.02
> 2000|0.98 ± 0.01|0.90 ± 0.04|0.0005 ± 0.0002|0.80 ± 0.10|0.67 ± 0.03|0.61 ± 0.02
>
> > **W4b** In Table 1, the improvement in Acc. (Due to Flood Complex) is negligible compared to Alpha+.
>
> ▶ Our Table 1 results show a small but *consistent* improvement over Alpha†, including on the new (requested by **b9wN**) `rocks` data (see table above).
> On closer inspection (via a paired t-test, as we evaluate on the *same* splits), the difference in mean between Flood PH and Alpha† PH is *statistically significant* (at 5%) on `swisscheese`, `MCB` and `rocks`, while on `corals` and `Modelnet10` it is not.
>
> Notably, for fair comparison, the #points used for Alpha† PH hinges on the runtime of Flood PH. In case of the latter, small implementation tweaks such as (1) allowing for larger batch sizes during Triton kernel computation (by transposing the computation grid), (2) implementing a Triton kernel for masking (see Sec. 4.2), and (3) adjusting block sizes for the GPU arch., yielded substantial runtime improvements (cf. Q1). Under these circumstances, the #points available to Alpha† decreases drastically and then **all** results clearly favor Flood PH (e.g., Alpha† on `corals` reduces to 0.63).
>
> > **W4c** Similarly, in Figure 3, it seems the accuracy of alpha(X) with 100k landmarks is almost the same as Flood complex with 500k landmarks...
>
> ▶ While it is true that the accuracy of Alpha(X) on 100k points is close to that of Flood(X,L), the latter is *much* faster (2 vs. 11s). One might expect better performance of Alpha PH with more points, but this is *not* necessarily the case. In fact, existing vectorization methods (including the one we use) suffer from *oversmoothing* effects as the number of points in the diagrams increases, resulting in less discriminative vectorizations and a drop in downstream performance. On `swisscheese` Alpha PH on all of $X$ yields 93% ± 2% (at 70s/point cloud) and on `MCB` 60% ± 3%  (at 200s).
>
> > **W4d** In Figure 4, explain why Alpha is slower than the Flood complex?
>
> ▶ Flood complex construction (1s) is faster than Alpha complex construction (90s) for *both* reasons you mention, and even a pure CPU-based implementation (which we offer in our code) is much faster (17s).
> The complex construction uses only a subset of the points, and the rest of the data is used only for the (computationally) cheap flooding process. Our GPU implementation of the latter then yields a notable speedup over the CPU-only variant.
>
> > Dim. 2 column in Fig. 4(b) is missing.
>
> ▶ Thank you for pointing this out; we will add the values to Fig. 4(b). For the Flood complex, runtime is 0.2s. For the Alpha complex, it is 1.3s (0.8s for construction and 0.5s for persistence).
>
> > **W5** Better clarification on the role of GPU in Flood complex construction...
>
> ▶ For landmark selection using FPS and the Delaunay triangulation of the landmarks, CPU-based implementations are efficient when using dedicated data structures such as $k$-d trees.
>
> In terms of runtime, the bottlenecks are (1) the pre-selection of relevant points per simplex and (2) the distance computations, see Sec. 4.2. Both steps are speed up when done on the GPU. That being said, we support a full ($k$-d tree based) CPU version (not submitted) to compute filtration values. For a typical case of 1M points and 2k landmarks in 3D, runtimes are around 17s, compared to 1s on GPU.
>
> ▶ **Q1** *That is correct*. PH computation time is negligible (usually, <0.1s), as PH is computed from a filtration of a comparably small complex, i.e., the Delaunay complex on the landmarks.
>
> > Could you provide a more detailed breakdown of the total time in a table?
>
> ▶ We report relative runtimes (in %) of landmark selection (FPS), complex construction, and PH computation on the *vertices* (ranging from 28k to 5.3M) of the coral meshes and on the `swisscheese` dataset (always 500k points) when using 1k and 2k landmarks. Avg runtimes are bout 1.3s (1k) and 1.8s (2k) on coral vertices,  and 0.7s and 1.2s on cheese.
>
> | |Corals (1k)|Corals (2k)|Cheese (1k)|Cheese (2k)|
> |-|-|-|-|-
> |FPS |6.8 ± 3.0|5.1 ± 2.3|11.3 ± 0.4|7.9 ± 0.2|
> |Complex|92.1 ± 2.6| 93.4 ± 2.0|86.3 ± 0.5|89.0 ± 0.2|
> |PH|1.0 ± 0.5|1.4 ± 0.5|2.3 ± 0.1|3.2 ± 0.1|
>
> When #landmarks is fixed, the share of time spent on complex construction increases with the point cloud size. When increasing #landmarks, the runtime of complex construction and PH computation increases more than that of FPS sampling.
>
> ▶ **Q2** The idea of a flooding process on a triangulation of a subset of a point cloud can be extended to different metrics in $R^n$, but guarantees (if there are any) are harder to show without the Voronoi-Delaunay duality. For (non-Euclidean) graphs, extending the Flood complex is difficult, as the flooding idea relies on distances between $X$ and points in the *convex hull* of vertices, and therefore on an ambient space.
>
> ▶ **Q3** As pointed out on `l198` of the paper, an upper bound is given by iterating, for each of the $m=O(|L|^{d/2})$ simplices, over the random points in $P_σ$ and the points in $X$, for a total of $O(m|P_σ||X|)$. With masking, one has $O(m|X|)$ distance computations to build the balls (see `l216`), and then for each simplex $O(|P_σ||\tilde X|)$, where $\tilde X\subset X$ is the set of points inside the masking ball. We will clarify this!

---

> > ### Comment · Reviewer_Sf3t · 2025-08-03
> >
> > > W1. Limited consideration of Baselines:
> >
> > -> Mostly resolved. One question regarding your experiments with [2] => Have you tried using  ripser(CPU) or ripser++ (GPU) instead of using gudhi for a faster computation?
> >
> > > W2. Sensitivity analysis:
> >
> > -> Slightly resolved. Few suggestions:
> > 1. Sometimes FPS is sensitive to the choice of initial node. Please consider reporting the std. by varying the initial node.
> > 2. This experimental result still does not show how landmark selection impacts the downstream ML task.
> >
> > > W3. Tradeoff between #landmarks and Approximation quality
> >
> > -> Not resolved. I was expecting a theoretical analysis connecting the # landmarks to the approximation quality in terms of d_B.
> >
> > > W4, W5.
> >
> > -> Resolved.
> >
> > > Some concern after reading Reviewer b9wN regarding Scope. (“I feel that the main contribution of this paper is in the field of computational geometry. I don't know how pertinent this work is to the NeurIPS audience.”)
> >
> > -> I share this concern. In particular, there are stronger models than Pointnet++ in the literature already (Pointnet++ is quite old, 2017), such as, (a) PointCNN (b) PointNeXt: Revisiting PointNet++ with Improved Training and Scaling Strategies (c) Point Transformer V3: Simpler, Faster, Stronger. Thus from a broader viewpoint, it is unclear how this progresses the field on Point cloud classification, segmentation etc.

---

> > > ### Author Response · Authors · 2025-08-04
> > >
> > > > W1
> > >
> > > ▶ Note that Ripser *does not* directly support witness complexes (but one could extract dist. matrices from GUDHI first). Nevertheless, most of the time is spent constructing the complex rather than computing PH. E.g., for only 100 landmarks and 10k witnesses, we already get ~10s for construction and <0.001s for PH. Thus, speeding up PH has almost no effect. The same reasoning applies (to a lesser degree) to [1].
> > > > W2
> > >
> > > ▶ Following your suggestion, we evaluated the effect of the initial node in FPS. While some spread is present, it does not affect our conclusions. We will include these results in the paper, thanks for pointing this out.
> > > Metric|Virus-FPS|Virus-Rand|Coral-FPS|Coral-Rand
> > > -|-|-|-|-
> > > $d_H(X,L)$|33.97±0.07|62.0±2.3|6.13±0.01|15.5±3.2
> > > $d_B-ℍ_0$|1.6±0.2|3.3±0.1|0.75±0.06|0.69±0.07
> > > $d_B-ℍ_1$|3.0±0.3|3.2±0.4|0.83±0.15|1.23±0.06
> > > $d_B-ℍ_2$|4.8±1.7|9.1±2.1|0.72±0.16|2.22±0.38
> > >
> > > For the impact of landmarks selection on downstream tasks, we ran experiments (FPS vs. uniform) on some datasets:
> > > **FPS**
> > > #Landmarks|cheese|corals|rocks (cls)|rocks (reg)
> > > -|-|-|-|-
> > > 20|0.81±0.03|0.53±0.12|0.56±0.07|$(6.8±1.9)×10^{-3}$
> > > 100|0.95±0.01|0.64±0.10|0.67±0.04|$(2.1±1.0)×10^{-3}$
> > > 2000|0.98±0.01|0.80±0.10|0.90±0.04|$(5.3±1.6)×10^{-4}$
> > >
> > > **Uniform sampling**
> > > #Landmarks|cheese|corals|rocks (cls)|rocks (reg)
> > > -|-|-|-|-
> > > 20|0.79±0.024|0.51±0.089|0.53±0.07|$(3.5±0.9)×10^{-3}$
> > > 100|0.93±0.027|0.76±0.101|0.64±0.06|$(1.5±0.2)×10^{-3}$
> > > 2000|0.99±0.008|0.81±0.084|0.89±0.04|$(5.7±1.5)×10^{-4}$
> > >
> > > Essentially, there is no clear *winner*. Yet, we argue that the ϵ-net property of FPS landmarks makes it preferable, especially when only few landmarks are used.
> > > > W3
> > >
> > > ▶ Sorry for the misunderstanding. We have some results in the paper, discussed in greater detail below:
> > >
> > > Recall that, in $ℝ^2$, our Cor. 4 provides an upper bound on $d_B$ based on the Hausdorff dist. $d_H(X,L)$. Essentially, it says that the approx. quality improves as the landmarks cover the point cloud more finely. This result is agnostic to the method used for selecting $L$ and we conjecture it to hold in any Euclidean space, see reply to **b9wN** (W10).
> > >
> > > A direct bound on the bottleneck dist. would be compelling, however, it requires strong assumption on the underlying point cloud and landmarks. Therefore, our and existing (tight!) approx. results for other complexes (e.g. for Rips and Cech) are typically in terms of the Hausdorff dist. between the two sets.
> > >
> > > For the same reasons, we cannot provide a *direct* guideline of *how many* landmarks are required. Still, when using FPS, the Hausdorff dist. between $L$ and $X$ decreases as $d_H(X,L)∈O(|L|^{-1/d})$, where $d$ is the intrinsic dimension, i.e., 2 for a surface (mesh) and 3 for a volume. See last reply to **b9wN** for a proof.
> > >
> > > Finally, we like to point out that for $L=X⊂ℝ^2$ we have an even stronger result than $d_B=0$, that is, $\mathrm{Flood}_r(X,X)$ and $\mathrm{Alpha}_r(X)$ are homotopy equiv. at any filtration value $r$.
> > > > Scope
> > >
> > > ▶ While our goal was not primarily to push the SOTA in point cloud classification, we believe our approach contributes to that subfield by providing (1) a new scalable TDA tool for practitioners, and (2) insights given by our experiments.
> > > We agree that there are more modern models than PointNet++, and on some datasets, these models indeed work very well (better than PH-based methods). *However*, in regimes where discriminative information is tied to *complex geometric small- and large-scale structures*, this is no longer the case. To show this point, we ran **PointMLP** [Ma et al., Rethinking network design and local geometry in point cloud: A simple residual MLP framework, In: ICLR '22] on our datasets, with results listed below. Preliminary results with yet another model, **PVT** [Zhang et al., PVT: Point-voxel transformer for point cloud learning, In IJII '22], look similar. Given the time constraints for this response, we couldn't run the suggested baselines yet, but will try to include them in a final version.
> > >
> > > ||cheese|corals|rocks (cls)|rocks (reg)|MCB|ModelNet10
> > > -|-|-|-|-|-|-
> > > PointNet++|0.52±0.03|0.55±0.10|0.58±0.06|(2.0±0.6)$×10^{-2}$|0.38±0.02|0.92±0.01
> > > PointMLP|0.52±0.04|0.55±0.12|0.54±0.05|(3.8±0.6)$×10^{-3}$|0.82±0.03|0.94±0.01
> > > PVT|0.51±0.03|0.59±0.10|0.50±0.03|(3.9±0.9)$×10^{-2}$|0.51±0.04|0.94±0.02
> > > **Ours**|0.98±0.01|0.80±0.10|0.90±0.04|(5.3±1.6)$×10^{-4}$|0.67±0.03|0.61±0.02
> > >
> > > **Importantly**, all tested architectures struggle on the structurally more challenging `cheese`, `rocks` and (real-world) `corals` datasets, which feature rich and complex geometric structures at multiple scales that are not easily learned by conventional or even recent point cloud arch., but can be captured by PH.
> > >
> > > In summary, we think enabling PH comput. for point clouds *at scale* is a good fit for the TDA+ML subfield (as acknowledged by **b9wN** in the rebuttal response), with implications for other ML domains where TDA approaches are helpful, such as robotics, 3D vision, or generative modeling.

---

> > > > ### Comment · Reviewer_Sf3t · 2025-08-05
> > > >
> > > > I thank the authors for their effort in presenting additional experiments and clarification. Almost all of my concerns are resolved.
> > > >
> > > > Although the lack of sound theoretical guidelines for #landmarks selection and the limitation to Euclidean space are limitations, the empirical results are compelling enough to make a case for the Flood Complex for point cloud data. Thus, I have adjusted my score to borderline accept.
> > > >
> > > > I understand that the limitation to Euclidean space is inherent by design, but regarding the first limitations, I do think some theoretical guideline in an asymptotic sense (even under some assumptions on the point cloud and landmark) has a significant merit. Provided the authors address this limitation to some extent, I am willing to further adjust my score.

---

> > > > > ### Author Response · Authors · 2025-08-07
> > > > >
> > > > > We sincerely thank you for your time and valuable feedback. We are glad that our responses could resolve your concerns, to the extent that you are even considering raising your score further.
> > > > >
> > > > > At this point, we are unable to provide a detailed theoretical analysis on the general case. We can, however, present some exemplary calculations to illustrate how our existing results can be used to assess the number of landmarks to be used. We also remark that, in most practical scenarios, the selection of the number of landmarks will be determined by computational constraints, possibly based on some validation data.
> > > > >
> > > > > Let us first assume a general setting where the topology of a point cloud $X$ is unknown. In $ℝ^2$, Cor. 4, i.e., $$d_B(\mathrm{dgm}_i(\mathrm{Flood}(X,L)), \mathrm{dgm}_i(\mathrm{Alpha}(X)) \le 3 d_H(X,L),$$ can be used together with a bound on the right-hand side. As described in our previous response, when using FPS, $d_H(X,L)$ scales with order $|L|^{-1/D}$ where $D$ is the intrinsic dimension of $X$. Conjecturing that Cor. 4 extends to higher ambient dimensions, it suffices to select $O(ε^{-D})$ landmarks.
> > > > >
> > > > > If $X$ is *more well-behaved*, then less landmarks will be necessary.
> > > > > For example, let $X = \lbrace x \in ℝ^2 \colon \|x\| = R\rbrace $ be a circle of radius $R$ and $L$ selected via FPS. We directly compute Flood PH, and we compare it to that of the thickenings $X_r = \bigcup_{x\in X} B_r(x)$. Diagrams for the latter are $\mathrm{dgm}_0(\lbrace X_r\rbrace _{r\ge 0}) = \lbrace (0,\infty)\rbrace $ and $\mathrm{dgm}_1(\lbrace X_r\rbrace _{r\ge 0}) = \lbrace (0,R)\rbrace $.
> > > > > The PH of $\mathrm{Flood}(X,L)$ is determined by the filtration values of the edges on the boundary of the triangulation (i.e., the ones "along the circle"). It can be shown that these edges have filtration value at most $\bar r = R(1-\cos(2\pi/|L|)) = O(R/L^2)$.
> > > > > Hence, at radius $\bar r$, the Flood complex becomes connected and the $ℍ_1$ feature is born. The only other topological change occurs at $r=R$, where the $ℍ_1$ feature dies.
> > > > > Consequently, $\mathrm{dgm}_0$ consists only of bars with death times at most $\bar r$, and $(0,\infty)$, and the bottleneck distance to $\mathrm{dgm}_0$ of $\lbrace X_r\rbrace _{r\ge 0}$ is at most $\bar r$ (matching all short bars to the diagonal). Similarly, the bottleneck distance between $ℍ_1$ diagrams is at most $\bar r$. This means that, in order to ensure bottleneck distances $<ε$, we need to select $|L|$ landmarks such that
> > > > > $$|L| \ge \frac{2\pi}{\arccos(1 - ε/R)} = O(\sqrt{R/ε}).$$
> > > > > It is worth mentioning that if one computes an Alpha complex on $L$, i.e., discarding the additional information available from $X$, then those edges have filtration value at most $R \sin(2\pi/L) = O(R/L)$, resulting in larger bottleneck distances, and one would need $ O(R/ε)$ landmarks (i.e., quadratically worse in $ε$). Presumably, this analysis extends to a general curvature-based argument in higher dimensions.
> > > > >
> > > > > This idealized example extends in multiple ways. First, if $X$ is deformed to a set $X'$ in a way that PH is preserved, e.g., to an annulus with the same inner radius, then Thm. 2 guarantees (in any dimension) that $$d_B(\mathrm{dgm}_i(\mathrm{Flood}(X',L)), \mathrm{dgm}_i(\lbrace X'_r\rbrace _{r\ge 0}) \le \bar r + d_H(X,X').$$
> > > > > If instead $X'$ is such that the PH is not preserved, e.g., if deformations are more general or if $X'$ is a finite subsample, then stability of the distance filtration ensures that
> > > > > $$d_B(\mathrm{dgm}_i(\lbrace X_r\rbrace _{r\ge 0})), \mathrm{dgm}_i(\lbrace X'_r\rbrace _{r\ge 0}) \le d_H(X,X')$$
> > > > > and therefore, under the assumption that the landmarks remain the same, that
> > > > > $$d_B(\mathrm{dgm}_i(\mathrm{Flood}(X',L)), \mathrm{dgm}_i(\lbrace X'_r\rbrace _{r\ge 0}) \le \bar r + 2 d_H(X,X').$$
> > > > > Since the second summand is constant in $|L|$, we still get the same asymptotic scaling in $|L|$.
> > > > >
> > > > > Finally, we give a toy example that exemplifies how to select the $|L|$ in a more complex scenario. Suppose that the point set consists of one small object of interest with diameter $R_1$ and a large object with diameter $R_2 \gg R_1$. We consider both the case in which they are surfaces (e.g., meshes) and solid objects. Assume that $k$ landmarks are needed to obtain an $ε$ approx. of the small object when analysing it in isolation. However, FPS or random selection of the landmarks will prefer the larger object and therefore the number of required landmarks will increase roughly by a factor $(R_2/R_1)^2$ (i.e., scaling with area) for surfaces/meshes, and $(R_2/R_1)^3$ (i.e., scaling with volume) for solid objects.
> > > > > In practice, these formulas can yield general guidelines on the number of landmarks needed on point clouds, given their diameter and the minimum scale of topological features of interest.
> > > > >
> > > > > Let us know if these derivations on some restricted point clouds are what you were looking for, and if you find them relevant enough to be included in the paper (appendix).

---

> > > > > > ### Comment · Reviewer_Sf3t · 2025-08-08
> > > > > >
> > > > > > Thank you for your response. I think the following part of our response, coupled with your response to Reviewer b9wN regarding "Choice of L" is to a large extent what I am looking for.
> > > > > >
> > > > > > > Let us first assume a general setting where the topology of a point cloud $X$ is unknown. In $ℝ^2$, Cor. 4, i.e., $$d_B(\mathrm{dgm}_i(\mathrm{Flood}(X,L)), \mathrm{dgm}_i(\mathrm{Alpha}(X)) \le 3 d_H(X,L),$$ can be used together with a bound on the right-hand side. As described in our previous response, when using FPS, $d_H(X,L)$ scales with order $|L|^{-1/D}$ where $D$ is the intrinsic dimension of $X$. Conjecturing that Cor. 4 extends to higher ambient dimensions, it suffices to select $O(ε^{-D})$ landmarks.
> > > > > >
> > > > > > - Am I correct to understand it as the following? => **Assuming that $L$ is an $\epsilon$-net of $X \in \mathbb{R}^2$, if $d_B(\mathrm{dgm}_i(\mathrm{Flood}(X,L)), \mathrm{dgm}_i(\mathrm{Alpha}(X)) \leq 3d_H(X,L) \leq 3\epsilon$, it is sufficient to have $|L| \geq \mathcal{O}(\epsilon^{-D})$**
> > > > > >
> > > > > > - The examples in your response are good to have in the supplementary material, but I think a succinct statement (similar in nature to what I stated) that connects $d_B$ bound with bound for \#landmarks would be a vital addition to the paper.

---

> ### Comment · Reviewer_Sf3t · 2025-08-08
>
> A small remark:
> > We also remark that, in most practical scenarios, the selection of the number of landmarks will be determined by computational constraints, possibly based on some validation data
>
> Such reasoning is problematic. One may also argue the following:
>
> "If I know that I can get $\epsilon$ guarantee in bottleneck distance by choosing, say, 10 landmarks (and if that is sufficent for my task), why would I waste computational resources by computing Flood complex with 1000 landmarks just because I can?"

---

> > ### Author Response · Authors · 2025-08-08
> >
> > Thanks for your additional comments.
> >
> > > Am I correct to understand it as the following?
> >
> > Almost. We would phrase it like this: If one selects $|L| > O(ε^{-D})$ via FPS, we have that $L$ is an $ε$-net of $X \subset \mathbb R^2$ and thus $d_B(\mathrm{dmg}_i (\mathrm{Flood}(X,L)), \mathrm{dmg}_i (\mathrm{Alpha}(X))) \le 3 d_H(X, L) \le  3 ε$.
> >
> > > but I think a succinct statement ... would be a vital addition to the paper.
> >
> > Absolutely. We will add such a statement directly after the approximation bound (Cor. 4) in the theory section (coupled with the essence of our response to Reviewer b9wN as you suggest), and we will include the examples in the supplementary material.

---

> ### Comment · Reviewer_Sf3t · 2025-08-08
>
> I thank the authors for their efforts during the rebuttal and discussion phase. Since I do not have any more concerns left, I have raised my score to Accept.
>
> I encourage the authors to incorporate the theoretical and empirical results, discussions, and missing related works into the final version. Good luck!

---

### Official Review · Reviewer_ruYk · 2025-07-01

**Clarity:** 4
**Significance:** 4
**Originality:** 1
**Rating:** 5
**Confidence:** 4

**Summary:**

This paper focuses on the central problem of the computational expense of persistent homology.  The paper proposes a new complex that is able to handle large scale point clouds and thus make the computation of persistent homology on some datasets that have so far not been possible.  Some theoretical guarantees are given and a nice, complete experimental study is carried out.

While I would not classify the contribution of the work to be under "AI", it does make persistent homology a viable tool to then be used to study problems in AI.

**Questions:**

Some comparisons are carried out wrt the alpha and witness complexes, for example, which I suppose provides a satisfactory (but somewhat underwhelming) "guarantee" of the performance of the method, but an important outstanding question that does not seem to be addressed is the correctness of the result, more specifically, the correctness of the algorithm (which was not detailed in the submission, I imagine that it probably appeared the appendix which was not submitted).  In the experiments, persistent homology was computed on massive datasets where it was to date not possible: how can we be sure that the resulting barcodes are indeed the correct barcodes?  If correctness cannot be proven, then some theoretical results on convergence of the method to the "true" barcode is a really important missing theoretical result.  Are there details on this in the missing appendix?  If so, they should be moved to the main body of the paper.

**Ethical Concerns:**

["NO or VERY MINOR ethics concerns only"]

**Final Justification:**

As discussed with the authors, although I do not find the approach original (especially after reading the rebuttal and realizing that their proposed new complex is homotopy equivalent to a very classical existing one), I do believe that the computational advantages offered are important and will have significant potential for impact in the field of topological data analysis. For these reasons, I have decreased the originality score but have increased my overall recommendation to accept the paper.

**Limitations:**

Limitations were discussed briefly in the main body of the paper, but given the potential impact of the work, I would like to see a more honest, sober analysis and ablation study of the contribution.

**Quality:**

4

**Strengths And Weaknesses:**

The problem that the paper studies is undoubtedly one of the most important problems in computational persistent homology and, in my view, one of the main limitations of the approach in applications: persistent homology simply does not scale well and there are serious problems with applying it to datasets that are common in current application areas and other fields, and definitely machine learning.  The contribution of this submission has the potential to overcome this and expand the potential and applicability of persistent homology as a tool.

A limitation of the work that I see is that it still relies on considering smaller subsets of the data, which is a natural/classical approach and takes away from the originality but nevertheless, the contribution of the work is definitely very important.  Additionally, proposing another complex is not very original, but again, nevertheless, a solid way to workaround the problem.  However, the biggest weakness I see and most problematic aspect is that it appears that the appendix was not submitted and so I could not check details (and would have liked to), which I see as quite a serious oversight.  This is the main reason for my borderline assessment of the paper.

---

> ### Author Rebuttal · Authors · 2025-07-30
>
> *We thank you for the valuable feedback and for recognizing the importance of our work.
> Below, we address all weaknesses (W) and questions (Q) point by point, enumerated as in the review. We specifically clarify the **availability** of the appendix as part of the supplementary material ZIP file.*
>
> > **Summary** While I would not classify the contribution of the work to be under "AI", it does make persistent homology a viable tool to then be used to study problems in AI.
>
> ▶ We understand your point, but respectfully disagree with the assertion that our work lies outside the (broad) domain of AI. In fact, we address (and remove) a key barrier to the adoption of TDA methods in modern AI/ML research, namely the inability to compute persistent homology at scale. By facilitating PH computation on millions of points, our approach allows the extraction of stable, interpretable features at practical scales for tasks where data comes with a complex geometric structure. The latter has direct implications for domains such as robotics and 3D vision, but also generative modeling (e.g., to ensure global structural fidelity) or biomedical AI (e.g., molecule analysis). From our perspective, Flood PH advances the computational and algorithmic foundations necessary to apply a theoretically rich tool (PH) in precisely these settings.
>
> > **W1** A limitation of the work that I see is that it still relies on considering smaller subsets of the data, which is a natural/classical approach and takes away from the originality but nevertheless, the contribution of the work is definitely very important.
>
> ▶ We want to clarify that our method consists of *two* building blocks: (1) the Delaunay complex on a *subset* of the data, i.e., the landmarks, and (2) a filtration on it that depends on the *entire* set of points. Contrary to simply considering only a subset of the point cloud (which is typically done in practice), in our construction, *all* points contribute to the filtration values. This is beneficial as it retains information that would otherwise be lost. On the `swisscheese` data (Appendix, Table 2), for example, we very clearly see that Flood PH allows to distinguish the two classes, while (even) repeated subsampling and computing Alpha PH fails to extract class-specific discriminative information due to insufficient subsample size.
>
> > **W2** Additionally, proposing another complex is not very original, but again, nevertheless, a solid way to workaround the problem.
>
> ▶ We understand the criticism. However, we observed that the existing complexes, even when paired with efficient/optimized PH computation (such as the ones provided by Gudhi or Ripser variants), are unable to compute PH beyond dimension 1 for large-scale problems, or for datasets with thousands of point cloud instances, as common in ML tasks. Given the unfavorable computational complexity for computing PH on large (in terms of #simplices) filtered simplicial complexes, the only viable solution is to introduce a new complex that is (1) small enough (achieved via the landmarks) to precisely control computational complexity of subsequent PH computation, and (2) amenable to construction on dedicated GPU-computing hardware. The proposed Flood complex realizes both of these points.
>
> > **W3** However, the biggest weakness I see and most problematic aspect is that it appears that the appendix was not submitted and so I could not check details (and would have liked to), which I see as quite a serious oversight. This is the main reason for my borderline assessment of the paper.
>
> ▶ The appendix (as well as the code) is available in the supplementary material (i.e., a ZIP file in our case), downloadable at the top of this page. Therein, we provide proofs for *all* theoretical statements, as well as additional details and experimental results.
>
> > **Q1** Some comparisons are carried out wrt the alpha and witness complexes, for example, which I suppose provides a satisfactory (but somewhat underwhelming) "guarantee" of the performance of the method,
>
> ▶ We primarily present experiments with the Alpha complex (including subsampling experiments in the Appendix). From a computational perspective, Alpha PH is the only viable competitor given the point cloud sizes in our experiments. Even publicly-available Witness complex implementations (e.g., in Gudhi) cannot be computed at that scale unless the number of landmarks is extremely low (around 100); see tables in our reply to reviewer **Sf3t**.
>
> > **Q2** ... an important outstanding question ... is the correctness of the algorithm. In the experiments, persistent homology was computed on massive datasets where it was to date not possible: how can we be sure that the resulting barcodes are indeed the correct barcodes? If correctness cannot be proven, then some theoretical results on convergence of the method to the "true" barcode is a really important missing theoretical result. Are there details on this in the missing appendix? If so, they should be moved to the main body of the paper.
>
> ▶ In Secs. 3.2 and 4.1, we present several approximation guarantees for our method. Specifically, we show (Theorem 3) in $ℝ^2$, that the Flood complex is topologically (homotopy) equivalent to the Alpha complex, if the landmark set $L$ is the entire point cloud $X$. Moreover, if $L\subset X$, then the (bottleneck) distance between the persistent homology of the Alpha and Flood complex is bounded by the Hausdorff distance between $X$ and $L$, see Cor. 4. We hope that our clarification on the availability of the appendix (and all proofs therein) as part of the supplementary material ZIP file clarifies this issue.
>
> > **Limitations** were discussed briefly in the main body of the paper, but given the potential impact of the work, I would like to see a more honest, sober analysis and ablation study of the contribution.
>
> ▶ In Appendix A.1., we specifically address limitations. We will (1) move the key elements of this section (e.g., homotopy equivalence only proven in $ℝ^2$) to the main part of the manuscript and (2) extend it by consolidating our sensitivity analysis - requested by multiple reviewers - to provide the reader with a clear picture of how the Flood complex behaves with varying parameters.

---

> > ### Comment · Reviewer_ruYk · 2025-08-01
> > **Rebuttal Acknowledgment**
> >
> > Thank you for detailed and thorough responses to my questions and concerns.  I now see how to access the supplementary material that I didn't see during the initial reviewing period; I will go through this and your responses to my questions in the coming days and get back to you as soon as I can.

---

> > > ### Comment · Reviewer_ruYk · 2025-08-06
> > > **Responding to authors and supplementary material**
> > >
> > > Thank you for responding to my comments in my original review. I have now read through the supplementary material to follow up on the concerns I had in my original review; I have also read the other reviews and corresponding author responses.
> > >
> > > Most of my concerns that are addressable have been mostly resolved; I do believe that an expanded sensitivity analysis and incorporation of limitations into the main contribution rather than in the supplementary materials is an important change that needs to be made.
> > >
> > > Although still have reservations about the originality of the proposed approach as I mentioned in my original review (and to which the authors responded), especially since the theoretical guarantees basically show homotopy equivalence to a classical and well-known other complex in the field, nevertheless, the computational advantage offered by the method will no doubt be impactful in TDA and I believe that it will open the door to further use of persistent homology in theoretical machine learning questions where it has already proven useful in some instances. I am happy to increase my score and recommend acceptance of the paper.

---

> > > > ### Author Response · Authors · 2025-08-07
> > > >
> > > > We sincerely thank you for your comments, which will surely help us improve our paper. We will make sure to include the expanded sensitivity analysis and to discuss limitations in the main paper.

---

### Official Review · Reviewer_kVdM · 2025-07-02

**Clarity:** 3
**Significance:** 3
**Originality:** 3
**Rating:** 4
**Confidence:** 1

**Summary:**

This paper introduces a novel construction called the Flood Complex, designed for efficient computation of PH on large-scale Euclidean point clouds. The key idea is to construct a filtered simplicial complex over a small set of landmark points by checking whether the convex hulls of Delaunay simplices are fully “flooded” (i.e., covered) by balls centered at the full dataset. This construction approximates the Alpha complex, while being orders of magnitude faster and scalable to millions of points. The authors provide theoretical guarantees (stability, approximation bounds), GPU-accelerated implementation, and extensive experiments showing state-of-the-art performance on both synthetic and real-world classification tasks.

**Questions:**

Please see my comments in the part of Strengths and Weaknesses.

**Ethical Concerns:**

["NO or VERY MINOR ethics concerns only"]

**Final Justification:**

All my concerns are resolved. Hence, I recommend acceptance.

**Limitations:**

Maybe yes.

**Quality:**

3

**Strengths And Weaknesses:**

Strengths:
+ The Flood complex is a creative and non-trivial extension of Alpha/Witness complexes, enabling scalable topological analysis on massive point clouds.
+ It tackles one of the central bottlenecks in Topological Data Analysis: applying PH at scale.

Weaknesses:
- In fact, I am completely unfamiliar with this direction. So, I don't know if it has any disadvantages.

---

> ### Author Rebuttal · Authors · 2025-07-30
>
> *Thank you for your overall positive feedback and recognizing the creativity and the relevance of our work.*
>
> > Weaknesses:
> In fact, I am completely unfamiliar with this direction. So, I don't know if it has any disadvantages.
>
> ▶ We understand that the paper is somewhat technical in parts. Yet, your summary very well captures the essence of our work, i.e., computing PH for low-dimensional Euclidean point cloud data at scale. We agree that this is, at the moment, one of the key limitations for ML practitioners when seeking to distill topological information out of data. Our approach is a first step towards mitigating this limitation in the low-dimensional regime.
>
> Further, thank you for giving your best effort. We appreciate your perspective and, aside from technicalities, we would be grateful if you would point us to any potentially unclear points so that we can make our work more accessible to a broader audience (outside the topological and geometric deep learning communities).

---

> > ### Comment · Reviewer_kVdM · 2025-08-05
> >
> > I have no any questions and will keep my original rating.

---

### Decision · Program_Chairs · 2025-09-17

**Decision:**

Accept (poster)

**Comment:**

This paper addresses the problem of the computational cost of computing the persistent homology. To overcome the issue (i.e., computing PH is general is expensive), the authors of the paper propose a new complex that is able to handle large scale point clouds and demonstrate its usefulness by computing the PH using their method on some datasets that have so far not been possible. The theoretical guarantees are supported by comprehensive experimental evaluations.

The paper is technically very strong and its contribution is believed to be impactful to the TDA community. However this is its weakness as well, as it's not clear how this is connected to the NeurIPS community. But as there have been TDA papers published at NeurIPS, this seems to be a minor issue.

All the reviewers agree that this is a strong paper with nice technical contributions. Therefore, it would be beneficial for the research community to have this paper presented at the conference.